# Unsupervised 3D Scene Representation Learning via Movable Object Inference

**Honglin Chen**[*1], **Wanhee Lee**[*1], **Hong-Xing Yu**[1], **Rahul Venkatesh**[1], **Joshua B. Tenenbaum**[2], **Daniel M. Bear**[1], **Jiajun Wu**[1], **Daniel L. K. Yamins**[1]

[1]*Stanford University*    [2]*MIT*    [*]*Contributed equally*

**Reviewed on OpenReview:** `https://openreview.net/forum?id=1QjCzPOKIw`

## Abstract

Unsupervised, category-agnostic, object-centric 3D representation learning for complex scenes remains an open problem in computer vision. While a few recent methods can discover 3D objects from a single image, they remain struggling on scenes with diverse and complex object configurations as they discover objects mostly by appearance similarity, which is insufficient for textured objects. In this work, we propose Movable Object Radiance Fields (MORF), aiming at scaling to complex scenes with diverse categories of objects. Inspired by cognitive science studies of object learning in babies, MORF learns 3D object representations via movable object inference. While obtaining 3D movable object signals requires multi-view videos of moving objects, we propose lifting a 2D movable object inference module that can be unsupervisedly pretrained on monocular videos. Thus, MORF requires only multi-view images of static training scenes. During testing, MORF can discover, reconstruct, and move unseen objects from novel categories, all from a single image of novel scenes. We propose a challenging simulated dataset with a diverse set of textured objects for training and testing. Experiments show that MORF extracts accurate object geometry and supports realistic object and scene reconstruction and editing, significantly outperforming the state-of-the-art.

## 1  Introduction

Learning object-centric 3D representations of complex scenes is a critical precursor to a wide range of application domains in vision, robotics, and graphics. The ability to factorize a scene into objects provides the flexibility of querying the properties of individual objects, which greatly facilitates downstream tasks such as visual reasoning, visual dynamics prediction, manipulation, and scene editing. Furthermore, we hypothesize that building factorized representations provides a strong inductive bias for compositional generalization (Greff et al., 2020), which enables understanding novel scenes with unseen objects and configurations.

While supervised learning methods (Ost et al., 2021; Kundu et al., 2022; Müller et al., 2022) have shown promise in learning 3D object representations (such as neural radiance fields (Mildenhall et al., 2020)) from images, they rely on annotations of specific object and scene categories. A recent line of work (Yu et al., 2022; Stelzner et al., 2021) has explored the problem of unsupervised discovery of object radiance fields. These models can be trained from multi-view RGB or RGB-D images to learn object-centric 3D scene decomposition without annotations of object segments and categories. However, these methods remain struggling with textured objects and objects from diverse categories. A fundamental reason is that they heavily rely on self-similarity of visual appearance to discover object entities, which limits their scalability beyond simple texture-less objects with similar shapes.

In this work, we aim to scale unsupervised 3D object-centric representation learning to complex visual scenes with textured objects from diverse categories. To this end, we propose Movable Object Radiance

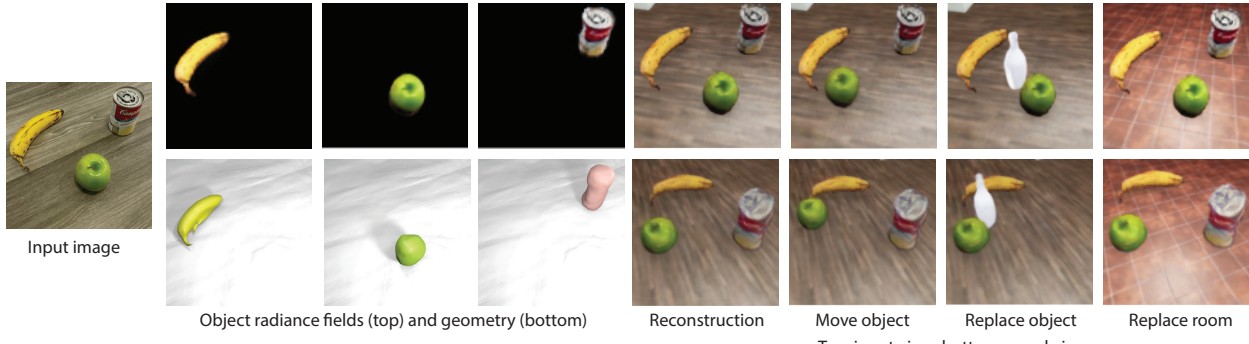

Figure 1: Illustration of unsupervised category-agnostic object-centric 3D representation learning. Given a single image (an unseen real image here, although our model is trained on synthetic images), we infer object radiance fields that allow photometric and geometric 3D reconstruction. This factorized representation enables 3D scene manipulation, including moving objects, replacing objects, and replacing the background.

Fields (MORF), which learns to infer 3D object radiance fields from a single image. Rather than appearance similarity, MORF uses material coherence under everyday physical actions as its underlying object definition, i.e., an object is movable as a whole in 3D space (Spelke, 1990). However, it is challenging to obtain learning signals to directly infer movable objects in 3D, which requires multi-view videos of moving objects. To address this challenge, we propose to lift 2D movable object discovery to 3D. MORF integrates a pretrained unsupervised 2D movable object segmentation module with differentiable neural rendering to bridge the gap between 2D segmentation and 3D movable object representations. We show that our lifting design allows learning 3D object-centric representations in complex category-agnostic scenes without multi-view videos.

Specifically, MORF integrates a recent unsupervised 2D movable object segmentation method, EISEN (Chen et al., 2022), which is pretrained on optical flow from unlabeled videos. The EISEN module is pretrained to segment objects in an image by perceptually grouping parts of a scene that would move as cohesive wholes. To lift the 2D movable object inference to 3D, MORF learns to condition 3D radiance field generation with 2D segmentations. We extract object-centric latent representations from segmented images and generate object radiance fields from the factorized latents. To allow high-quality reconstruction of textured objects, our learned latent object representation consists of both an entity-level latent and a pixel-level latent that better encodes spatially varying appearance details.

To evaluate our method, we propose a challenging simulated dataset *Playroom* with a diverse set of textured objects, going beyond scenes with simplistic object appearances considered by most current unsupervised 3D object discovery methods (Yu et al., 2022; Stelzner et al., 2021; Sajjadi et al., 2022b). We demonstrate that MORF can learn high-quality 3D object-centric representations in complex visual scenes, allowing photometric and geometric reconstruction for these scenes from single views. Moreover, our learned representations enable 3D scene manipulation tasks such as moving, rotating, replacing objects, and changing the background. Beyond systematic generalization to unseen spatial layouts and arrangements, we further show that MORF is able to generalize to unseen object categories and even real images while maintaining reasonable reconstruction and geometry estimation quality (Figure 1).

In summary, our contributions are three-fold. First, we propose scaling unsupervised object-centric 3D representation learning to scenes with textured objects from diverse categories by discovering objects with coherent motions. Second, we propose Movable Object Radiance Fields (MORF), which lifts 2D movable object discovery to 3D to remove the requirements of multi-view videos of dynamic training scenes. Third, we demonstrate that MORF allows photometric and geometric reconstruction and editing of complex 3D scenes with textured objects from diverse unseen categories on the *Playroom* dataset.

## 2 Related Work

**Unsupervised 2D object discovery.** Our method is closely related to recent work on unsupervised scene decomposition, which aims to decompose multi-object scenes into separate object-centric representations

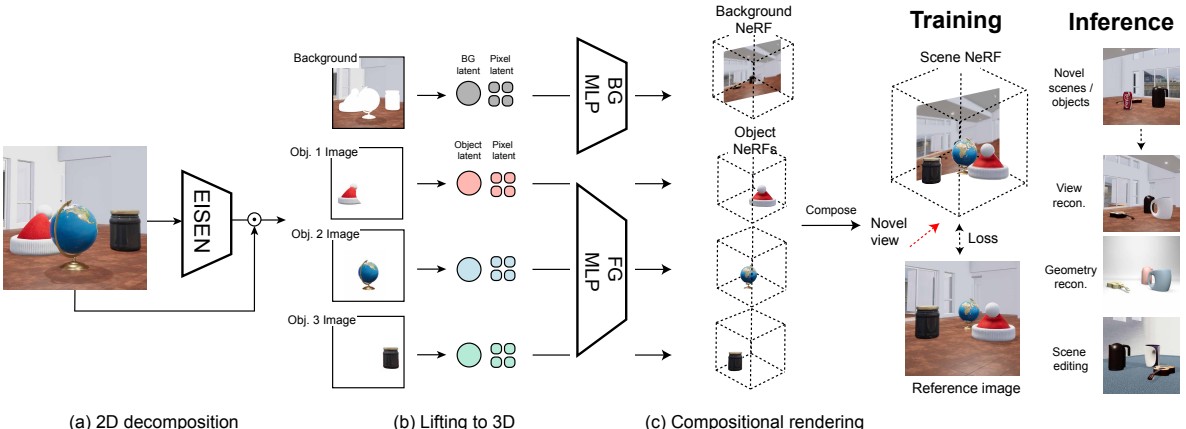

Figure 2: Illustration of Movable Object Radiance Fields (MORF). MORF takes as input a single image of a scene with potentially diverse objects, and infers 3D object and background radiance fields. (a) MORF integrates an image-based movable object inference method, EISEN, which infers a set of object masks. Masked images are used to generate object radiance fields. (b) MORF generates object radiance fields conditioned on the latent object and pixel codes. (c) During training, MORF reconstructs the novel view via compositional rendering and is supervised by reconstruction losses. During inference, MORF takes a single view of a new scene, and infers object and background radiance fields in a single forward pass.

from images without human annotations. Most works formulate the problem as learning compositional generative models in the 2D image space. They decompose a visual scene into a set of localized object-centric patches (Eslami et al., 2016; Crawford & Pineau, 2019; Kosiorek et al., 2018; Lin et al., 2020; Jiang et al., 2019a) or a set of scene mixture components (Burgess et al., 2019; Greff et al., 2019; 2016; 2017; Engelcke et al., 2019; Locatello et al., 2020; Monnier et al., 2021; Jiang et al., 2019b). The scene mixture models typically generate single-object RGBA images and blend them to reconstruct the full-scene images using iterative inference with recurrent networks (Burgess et al., 2019) or set-based convolutional encoders (Locatello et al., 2020). However, these methods have so far been unable to scale to complex real-world images. A recent branch of work on self-supervised or unsupervised object segmentations explores additional supervision signals such as motions and depth for learning object segmentations (Bear et al., 2020; Kipf et al., 2021; Chen et al., 2022; Bao et al., 2022; Elsayed et al., 2022; Ye et al., 2022). However, these 2D methods are not aware of the 3D nature of scenes, and thus they do not provide a 3D understanding of the underlying scenes.

**Unsupervised 3D object discovery.** Discovering objects from image collections has been a long-standing topic in computer vision, but previous work on object discovery (a.k.a. co-segmentation) represents objects as 2D image segments without 3D information (Russell et al., 2006; Sivic et al., 2005; 2008; Grauman & Darrell, 2006; Joulin et al., 2010; Rubio et al., 2012; Vicente et al., 2011; Rubinstein et al., 2013; Cho et al., 2015; Li et al., 2019; Vo et al., 2020). Recently, some attempts have been made to discover 3D object representations. A related set of methods focuses on 3D reconstruction from a single image (Ye et al., 2021; Kulkarni et al., 2019; 2020; Kanazawa et al., 2018; Wu et al., 2021). However, it requires strong category-specific shape priors, making it difficult to scale to complex real-world data. Elich et al. (Elich et al., 2020) infer object shapes (Park et al., 2019) from a single image of a scene. Chen et al. (Chen et al., 2020) extend Generative Query Network (Eslami et al., 2018) to decompose 3D scenes. In particular, the closest to our work are those that focus on inferring 3D neural object representations from single images (Yu et al., 2022; Smith et al., 2022; Stelzner et al., 2021) or sparse views (Sajjadi et al., 2022b). However, these methods rely on visual appearance to discover object entities. This fundamental assumption makes it difficult for them to scale to complex scenes with textured objects, diverse object categories, or objects under different lighting. In contrast, our approach leverages motions as the underlying object concept, which is category-agnostic and generalizable to different object appearances.

**Scene de-rendering.** Aiming to provide full scene understanding, a line of scene de-rendering works have shown reconstructing 3D object-centric representations in specific types of scenes (Wu et al., 2017; Yao

et al., 2018; Kundu et al., 2018; Ost et al., 2021; Kundu et al., 2022; Yang et al., 2021; Wu et al., 2022). Recently, Ost et al. (Ost et al., 2021) propose to represent dynamic scenes as a scene graph, where each node encodes object-centric information. Muller et al. (Müller et al., 2022) recover 3D object information such as shape, appearance, and pose in autonomous driving scenes. However, these methods rely on manual annotations of object categories (such as cars) and scene categories (such as street scenes). Similarly, Gkioxari et al. (Gkioxari et al., 2022) propose a method that learns to predict the 3D shape and layout of 3D objects relying on the supervision of the 2D bounding box. Our approach only requires object motion in video to infer object segmentations during training, without requiring manual annotations.

**Neural scene representations and rendering.** Our object representation is based on recent progress in neural scene representations (Park et al., 2019; Mescheder et al., 2019; Sitzmann et al., 2019) and neural rendering (Tewari et al., 2020). Neural scene representations implicitly model 3D scenes using the parameters of deep networks, which could be learned from only 2D images (Niemeyer et al., 2020; Sitzmann et al., 2019) with differentiable rendering techniques (Kato et al., 2020; Tewari et al., 2020). Specifically, Neural Radiance Fields (NeRFs) (Mildenhall et al., 2020) have shown photorealistic scene modeling of static scenes using only images. The most relevant works in this line aim to infer NeRFs from a single image (Yu et al., 2020; Kosiorek et al., 2021; Rematas et al., 2021). While these works focus on single objects or holistic scenes, we address decomposing a multi-object scene without human supervision. Another relevant line of work aims to incorporate NeRFs into compositional models (Niemeyer & Geiger, 2020; 2021), such as GIRAFFE (Niemeyer & Geiger, 2020). While they target scene synthesis, we instead focus on multi-object inference, which GIRAFFE cannot address (Yu et al., 2022).

# 3 Movable Object Radiance Fields (MORF)

We now describe the problem formulation and our approach, Movable Object Radiance Fields. Given a single input image of a scene that potentially has objects from diverse categories, our goal is to factorize the scene into a set of object-centric radiance fields. To allow unsupervised decomposition of such complex 3D scenes, we propose learning to discover 3D movable objects by integrating and lifting unsupervised 2D movable object inference. The lifting is done by learning to condition 3D radiance fields with 2D object segments using neural rendering.

Movable Object Radiance Fields has three stages. First, it decomposes the input image by inferring its 2D object segmentation masks (Figure 2a). The segmentation extraction network is trained with only optical flow to group scene elements that often move together into an object segment. Second, for each mask, MORF learns an object radiance field with object and pixel latent codes for object features and locally varying details, respectively (Figure 2b). Finally, these object radiance fields are then composed to re-render the scene from multiple views, supervised by the reconstruction loss (Figure 2c). We now describe each component in detail.

## 3.1 Movable object inference in 2D

Given a single RGB image, we first compute a 2D object segmentation mask, represented as a set of $H \times W$ dimensional binary-valued tensor $\{\mathbf{M}_i\}_{i=1}^{K}$, where $K$ is the number of object masks. We denote the background mask as $\mathbf{M}_0$ which can be computed by taking the complement of the object masks union: $\mathbf{M}_0 = (\bigcup_{i=1}^{K} \mathbf{M}_i)^c$.

We adopt the EISEN architecture (Chen et al., 2022) to generate high-quality object segmentation masks. The core idea of EISEN is to construct a high-dimensional plateau map representation (of shape $H' \times W' \times Q$) for each image, in which all feature vectors belonging to the same object are aligned (i.e., have cosine similarity $\approx 1$) and all feature vectors belonging to other objects are nearly orthogonal (cosine similarity $\approx 0$). Given this representation, the object segments can be easily extracted from a plateau map by finding clusters of vectors pointing in similar directions.

More specifically, EISEN first applies a convolutional backbone to the input image to obtain a feature grid, followed by an affinity prediction module that computes pairwise affinities between the features at pixel

location $(i, j)$ and the features at its neighboring locations within a local window. Then EISEN constructs a graph, with nodes represented by $Q$-dimensional feature vectors and edges represented by pairwise affinities. A message-passing graph neural network is run on the graph to construct the plateau map representations, passing excitatory messages that align the feature vectors of nodes belonging to the same object and inhibitory messages that orthogonalize the feature vectors of nodes belonging to distinct objects. Once the plateau map representation is obtained, EISEN imposes winner-take-all dynamics on the plateau map to extract object segments. We refer the readers to Chen et al. (2022) for more implementation details of EISEN.

During training, EISEN learns pairwise affinities from optical flow estimates from a RAFT (Teed & Deng, 2020) network pretrained on Sintel (Mayer et al., 2016). Consider a pair of scene elements $(a, b)$ that project into image coordinates $(i, j)$ and $(i', j')$ respectively. If only one is moving, it is usually the case that they do not belong to the same object; when neither is moving, there is no information about their connectivity. We use this physical logic to construct a pairwise supervision signal for EISEN's affinity matrix. The EISEN training loss is the masked row-wise KL divergence between the predicted and the target connectivity. Although EISEN requires a frame pair as input for computing optical flow during training, it only requires a single static image during inference time for computing the object segmentation masks, and subsequently predicting the object radiance fields.

### 3.2 Lifting by learning to condition radiance fields

We represent 3D objects and the background in a scene as neural radiance fields (NeRFs). We incorporate and lift the 2D movable object inference by learning to condition the NeRFs with the 2D segments. To allow high-fidelity reconstruction of complex scenes with spatially varying appearances for objects, each conditioning latent feature consists of both an entity-level slot-based latent and a pixel-level local latent, as shown in Figure 2b. In the following, we elaborate on the latent representations.

**Object-level latents.** Given an input image $\mathbf{I}$ and object masks $\{\mathbf{M}_i\}_{i=0}^K$ inferred by a frozen pre-trained EISEN, we extract a set of object latents (where $i = 0$ denotes the background) $\{\mathbf{s}_i\}_{i=0}^K$, each of which represents object-level features such as shape and position. We denote a masked image through $\mathbf{M}_i$ as $\mathbf{I}_i$, which is visualized on the left of Figure 2b. We extract an object latent $\mathbf{s}_i$ by a convolutional backbone encoder $E_s$ that produces a feature map $\mathbf{f}_i = E_s(\mathbf{I}_i)$, and average-pool it as $\mathbf{s}_i = \text{avgpool}(\mathbf{f}_i)$. We further allow the object latent to be adaptively updated by a learnable Gated Recurrent Unit (Chung et al., 2014) (GRU): $\mathbf{s}_i \leftarrow \text{GRU}(\mathbf{s}_i, \mathbf{u})$ for a few iterations, where $\mathbf{u}$ denotes an update signal given by aggregating features $\mathbf{f}_i^s$ that are within the object mask:

$$\mathbf{u} = \sum_{j=1}^{H \times W} w_j \cdot \mathbf{V}^T \mathbf{f}_i^{(j)}, \quad w_j = (\mathbf{M}_i^{(j)} / \sum_{j=1}^{H \times W} \mathbf{M}_i^{(j)}),$$

where $\mathbf{V}$ denotes a learnable linear projection matrix. Note that we use a shared GRU and a shared $\mathbf{V}$ for all objects, while we use separate ones for the background because of the inherent appearance and geometry differences between foreground objects and background.

**Pixel-level latents.** Object-level latents fall short of capturing spatially varying fine details. Thus, we also learn to extract pixel-level local latents $\mathbf{p}_i(\mathbf{x})$ for each object $i$ to condition the 3D object radiance field. To obtain the pixel latent codes, we use convolutional encoder $E_p$ that produces a feature map $\mathbf{g}_i = E_p(\mathbf{I}_i)$. For each 3D query point $\mathbf{x}$ on a camera ray, we follow Yu et al. (2020) to retrieve the image features of each object by projecting $\mathbf{x}$ onto the image plane coordinates $\pi(\mathbf{x})$ and extract latent codes $\mathbf{p}_i(\mathbf{x})$ from the feature map $\mathbf{g}_i$ by bilinear interpolation.

**Conditioning NeRFs.** Since the geometry of the background and foreground objects are inherently different, we parameterize the background NeRFs and object NeRFs using two separate conditional MLPs $f_b$ and $f_o$, respectively:

$$f_b : (\mathbf{x}, \mathbf{d}, \mathbf{s}, \mathbf{p}) \rightarrow (\sigma, \mathbf{c}), \quad f_o : (\mathbf{x}, \mathbf{d}, \mathbf{s}, \mathbf{p}) \rightarrow (\sigma, \mathbf{c}), \tag{1}$$

where $\mathbf{x}$ denotes a 3D position and $\mathbf{d}$ denotes a view direction. The MLP parameters of the object NeRFs are shared across all foreground objects.

### 3.3 Compositional neural rendering

At each point in the rendering of a scene, the final pixel value is a combination of contributions from each individual scene element to the 3D volumes projecting to that point. We follow uORF (Yu et al., 2022) and take a weighted average of the individual components to obtain the combined density $\sigma$ and color $\mathbf{c}$. The composite volumetric radiance field is then rendered into a 2D image via the numerical integration of volume rendering by sampling $S$ discrete points along each pixel ray parameterized as $\mathbf{r}(t) = \mathbf{o} + t\mathbf{d}$, with the ray origin $\mathbf{o}$ and view direction $\mathbf{d}$. The points on each ray are sampled between pre-specified depth bounds $[t_n, t_f]$, with distance $\delta_j$ between adjacent samples along the ray. Thus, the final pixel color is given by:

$$\mathbf{c} = \sum_{i=1}^{S} T_i[1 - \exp(-\sigma_i \delta_i)]\mathbf{c}_i, \quad T_i = \exp(-\sum_{j=1}^{i} \sigma_i \delta_i), \tag{2}$$

$$\sigma = \sum_j p_j \sigma_j, \ \mathbf{c} = \sum_j p_j \mathbf{c}_j, \ p_j = \frac{\sigma_j}{\sum_k \sigma_k}, \ j, k \in \{0, ..., K\}, \tag{3}$$

where we let $i = 0$ denote the background quantity, and similarly for $j = 0$ and $k = 0$.

### 3.4 Loss function

During training, we randomly select a single image of a scene as input and render multiple novel views. We train the model using both reconstruction loss and perceptual loss: $\mathcal{L} = \mathcal{L}_r + \lambda_p \mathcal{L}_p$ with $\lambda_p = 0.006$. The reconstruction loss is the L2 loss between the rendered image and the ground-truth image $\mathcal{L}_r = ||I - \hat{I}||^2$. Since reconstruction loss is sensitive to small geometric imperfections and often results in blurry reconstructions, especially due to uncertainties in rendering a novel view, we add a perceptual loss term to mitigate this problem. We compute the perceptual loss as $\mathcal{L}_p = ||e_k(I) - e_k(\hat{I})||^2$ , where $e_k(\cdot)$ is k-th layer of an off-the-shelf VGG16 (Simonyan & Zisserman, 2014) image encoder $e$ with frozen pre-trained weights.

## 4 Experiments

We demonstrate our approach on three tasks: (a) novel view synthesis, (b) scene geometry reconstruction, and (c) editing scenes by moving objects, replacing objects, and changing the background.

**Datasets**. We generated four variants of a complex synthetic scene dataset using the ThreeDWorld simulation environment (Gan et al., 2020). Each scene includes four camera views with a random azimuth angle and a fixed elevation; the camera always points to the center of the scene. In addition, we also train and evaluate our models on the *MultiShapeNet* dataset (Stelzner et al., 2021).

*Playroom* contains a wide range of realistically simulated and rendered objects. Each scene includes 3 objects randomly sampled from a set of 2000 object models. These models are drawn from a wide set of categories and have a range of complex 3D shapes and textures. They are placed in random positions and poses in rooms selected randomly from a collection of indoor environments with varying 3D room layouts and floor/wall textures. In each scene, one object is invisibly pushed to generate object motion. There are 15,000 scenes for training and 600 for testing.

*Playroom-novel* contains a mixture of both novel object instances from categories seen during training, as well as object instances from novel categories for evaluating the generalization performance of the models. Object instances from novel categories are randomly sampled from 36 held-out categories. We randomly sample 100 novel objects in total. They are placed in the same room environments seen in the *Playroom* dataset. There are 600 scenes for evaluation.

*Playroom-diverse* extends *Playroom* dataset to include more diverse background environments. *Playroom* dataset uses 3 environments, whereas *Playroom-diverse* contains 30 unique background environments. There are 30,000 additional scenes for training and 600 for testing.

| Models | Playroom | | | Playroom-novel | | | Playroom-diverse | | | MultiShapeNet | | |
|---|---|---|---|---|---|---|---|---|---|---|---|---|
| | LPIPS↓ | SSIM↑ | PSNR↑ | LPIPS↓ | SSIM↑ | PSNR↑ | LPIPS↓ | SSIM↑ | PSNR↑ | LPIPS↓ | SSIM↑ | PSNR↑ |
| uORF | 0.348 | 0.634 | 21.5 | 0.350 | 0.636 | 21.6 | 0.553 | 0.547 | 18.1 | 0.307 | 0.802 | 25.1 |
| pixelNeRF | 0.250 | 0.745 | 24.4 | 0.265 | 0.725 | **23.1** | 0.357 | 0.650 | 20.4 | 0.159 | **0.901** | **30.2** |
| SRT | 0.352 | 0.704 | 21.3 | 0.355 | 0.704 | 21.3 | 0.451 | 0.582 | 18.7 | 0.342 | 0.829 | 26.1 |
| MORF-nopix | 0.244 | 0.735 | 23.0 | 0.264 | 0.722 | 22.2 | 0.466 | 0.602 | 18.4 | 0.317 | 0.813 | 25.3 |
| MORF (ours) | **0.161** | **0.784** | **24.5** | **0.189** | **0.755** | 22.9 | **0.248** | **0.675** | **21.3** | **0.147** | 0.881 | 28.9 |

Table 1: Novel view synthesis performance measured by LPIPS, SSIM, and PSNR metrics on four datasets.

*Playroom-edit* is designed to evaluate a model's ability to manipulate object radiance fields and synthesize novel images. The dataset contains scenes that result from three types of editing: moving objects, replacing objects, and changing the background. For moving objects, we randomly pick one object in the scene and teleport it to a random position. For replacing objects, we switch a randomly selected object with an object from a different scene. For changing the background, we similarly switch the background with that of another scene. For each scene editing task, we render 200 test scenes for evaluation.

*MultiShapeNet* is a challenging multi-object dataset in which the objects are shapes from the ShapeNet dataset. Each scene contains 2–4 upright ShapeNet objects sampled from the chair, table, and cabinet classes, for a total of approximately 12,000 objects (Stelzner et al., 2021). We train the models on 80,000 training scenes and report the quantitative results on 500 validation scenes. We cannot train EISEN on the MSN dataset since there is no motion. We thus replace EISEN with a Mask2Former (Cheng et al., 2021) model pretrained on the COCO dataset to compute the 2D segmentations of the dataset.

**Baselines**. We compare MORF to the slot-conditioned unsupervised object radiance field method uORF (Yu et al., 2022) and nonfactorized pixel-conditioned method pixelNeRF (Yu et al., 2020). Both methods learn radiance fields from RGB images without ground-truth object annotations or depth supervision. We also compare to an ablated version of MORF trained without pixel latents to illustrate the benefits of local conditioning in 3D representation learning. We adopt the same training procedures and hyperparameter choices as reported in the original papers. For a fair comparison, all models receive input images with the same resolution and are trained with the same batch size.

## 4.1 Novel view synthesis

We randomly select one camera view of each scene as input and use the remaining three images as ground truth to evaluate the quality of novel view synthesis. In addition to uORF and pixelNeRF, we also compare with the pixel feature–ablated version of MORF. All models are evaluated using the standard image quality metrics PSNR, SSIM (Wang et al., 2004), and LPIPS (Zhang et al., 2018).

**Results**. As shown in Table 1 and Figure 3, MORF outperforms the baseline methods both quantitatively and qualitatively. Figure 3a and Figure 3b show the novel view synthesis results on *Playroom* and *Playroom-novel*. uORF and SRT (Sajjadi et al., 2022a) are able to learn rough object decomposition and position estimates, but fail to represent the object shapes accurately, resulting in inaccurate rendering from novel views. The version of MORF without pixel features as conditioning performs second best. Although both uORF and pixel feature–ablated MORF use object latent conditioning, the latter attains substantially better reconstructions, suggesting that accurate object segmentations help constrain the optimization of the neural implicit function. Both MORF and pixelNeRF are able to render novel views reasonably well. We highlight that pixelNeRF's reconstructions are blurry both in background regions (such as floor tiles) and on objects (such as the dumbbell), while MORF's reconstructions are sharper. Figure 3c shows the reconstructed novel views on *MultiShapeNet* dataset. Both uORF and the pixel feature-ablated MORF fail to decompose the scenes into multiple object slots, resulting in blurry reconstructions of the objects. This demonstrates the importance of accurate object decomposition in high-quality novel view synthesis. The comparison to the pixel feature-ablated MORF illustrates the advantage of using pixel features over object latents in capturing fine-grained details of scenes.

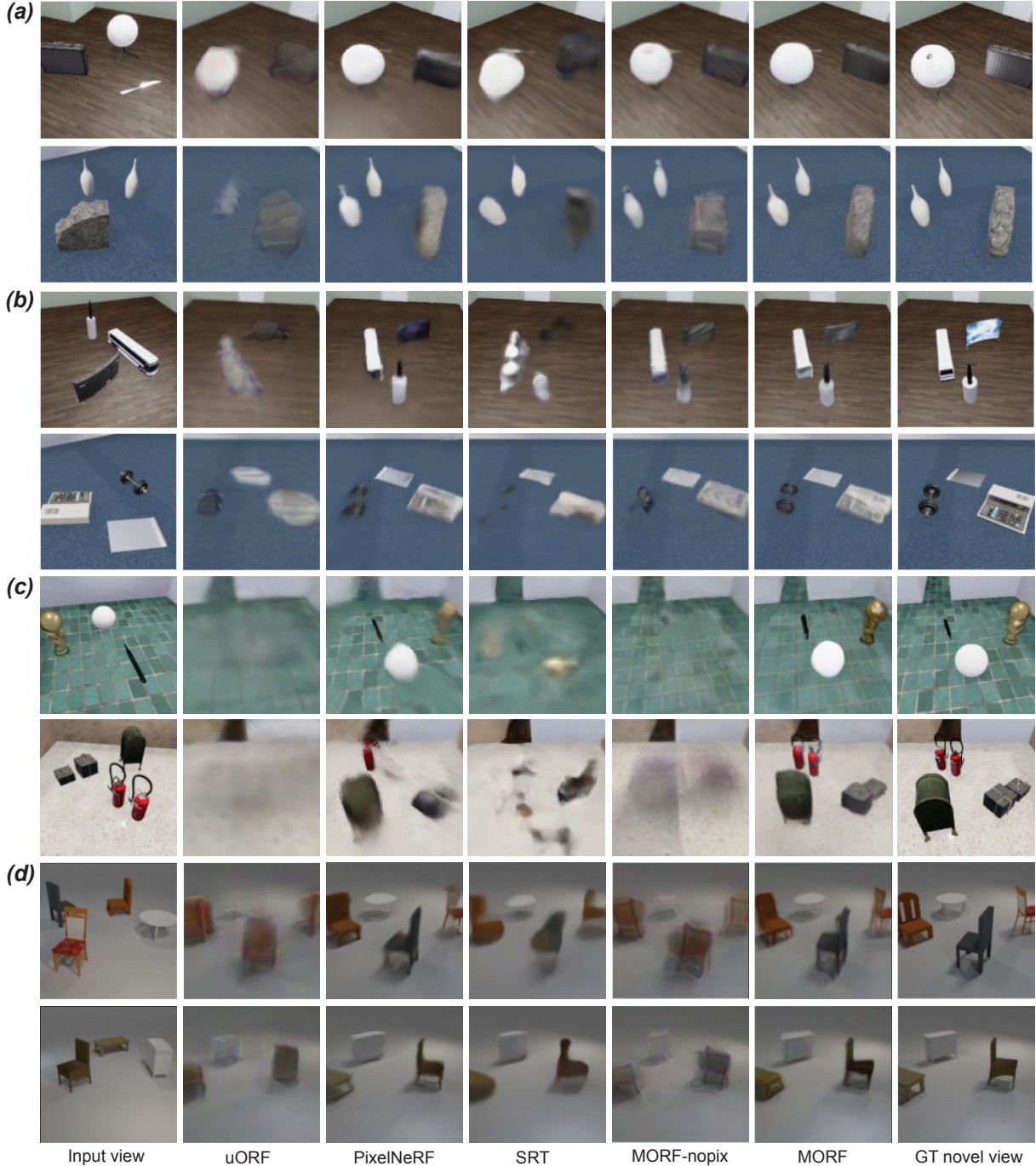

Figure 3: Novel view synthesis results on (a) *Playroom*, (b) *Playroom-novel*, (c) *Playroom-diverse*, and (d) *Multi-ShapeNet* datasets. The objects in *Playroom-novel* dataset are not included in the training set. MORF outperforms the baseline models on both foreground and background reconstruction. MORF is better at reconstructing fine-grained textures and object geometries than the other models.

## 4.2 Geometry reconstruction

To evaluate the quality of 3D scene representations, we evaluate both scene segmentations in 3D and extracted meshes from each model's learned density field. The segmentations of uORF and MORF are obtained by first volume-rendering density maps of objects and the background, and then assigning the pixel to the one with the highest density. To extract the meshes of each object, we first compute the density field by

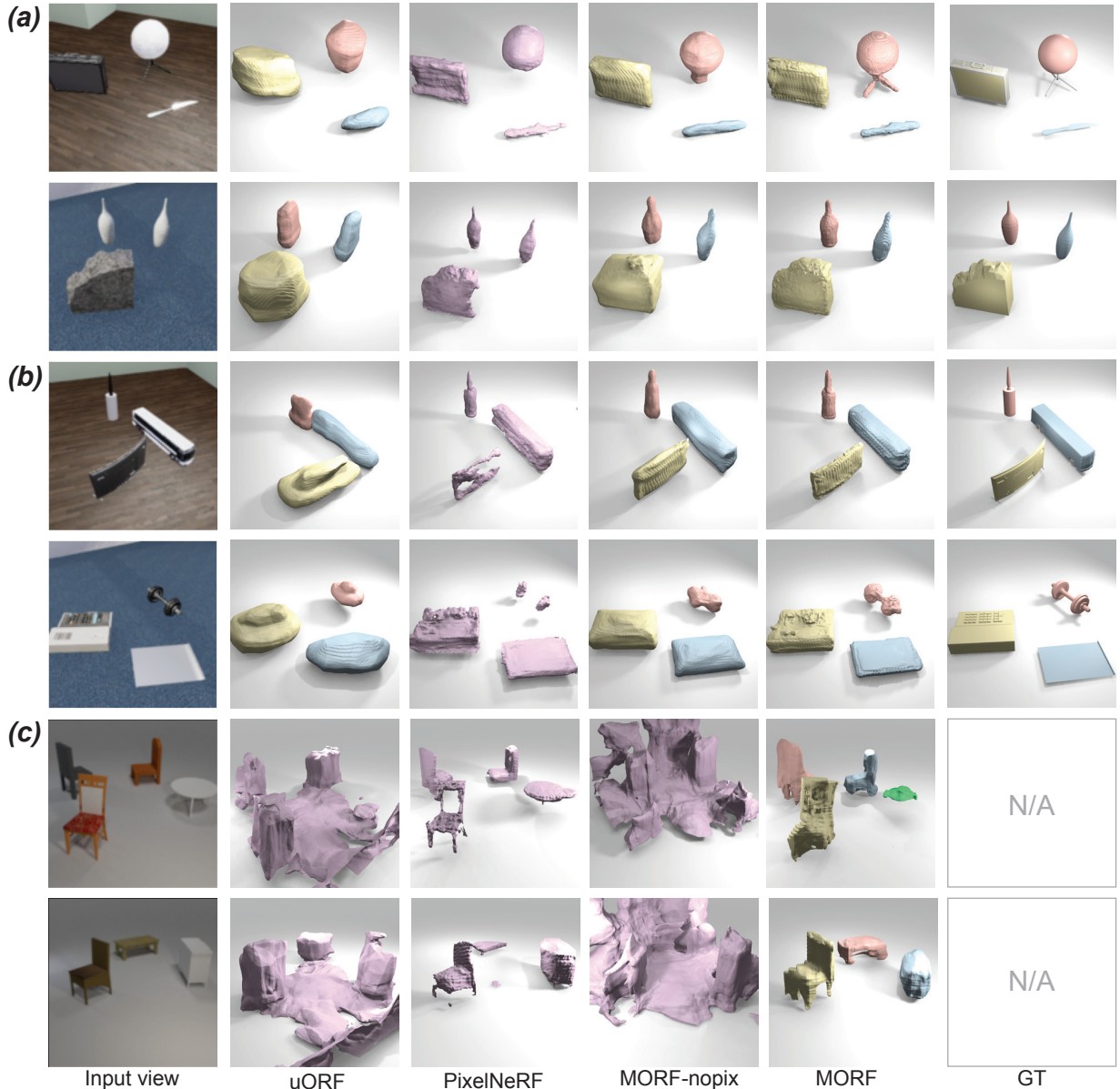

Figure 4: 3D mesh reconstruction results on (a) *Playroom*, (b) *Playroom-novel*, and (c) *MultiShapeNet* datasets. MORF produces more accurate mesh reconstructions than uORF and pixelNeRF, as well as the ablated model of MORF trained without pixel features as conditioning. PixelNeRF only outputs a single mesh encompassing both the foreground objects and the background. We remove the background mesh in pixelNeRF and visualize the foreground objects only. Mesh reconstruction for SRT is not applicable due to the lack of explicit geometry representation. The ground-truth meshes for *MultiShapeNet* are unavailable and therefore shown as N/A.

evaluating the foreground decoder at grid points in the world coordinate system, followed by the marching cubes algorithm (Lewiner et al., 2003). For a fair comparison, all models are evaluated using the same grid size of 256.

**Metrics**. To evaluate the quality of scene segmentations in 3D, we compute the mean Intersection over Union (mIoU) between the segmentations of a novel view and the ground-truth segmentations. In addition, we evaluate the quality of the meshes using Chamfer Distance (CD) (Sun et al., 2018). We compute two types of CDs, object mesh Chamfer Distance (O-CD) and scene mesh Chamfer Distance (S-CD). The O-CD metric measures the quality of mesh reconstruction for each individual foreground object, whereas the

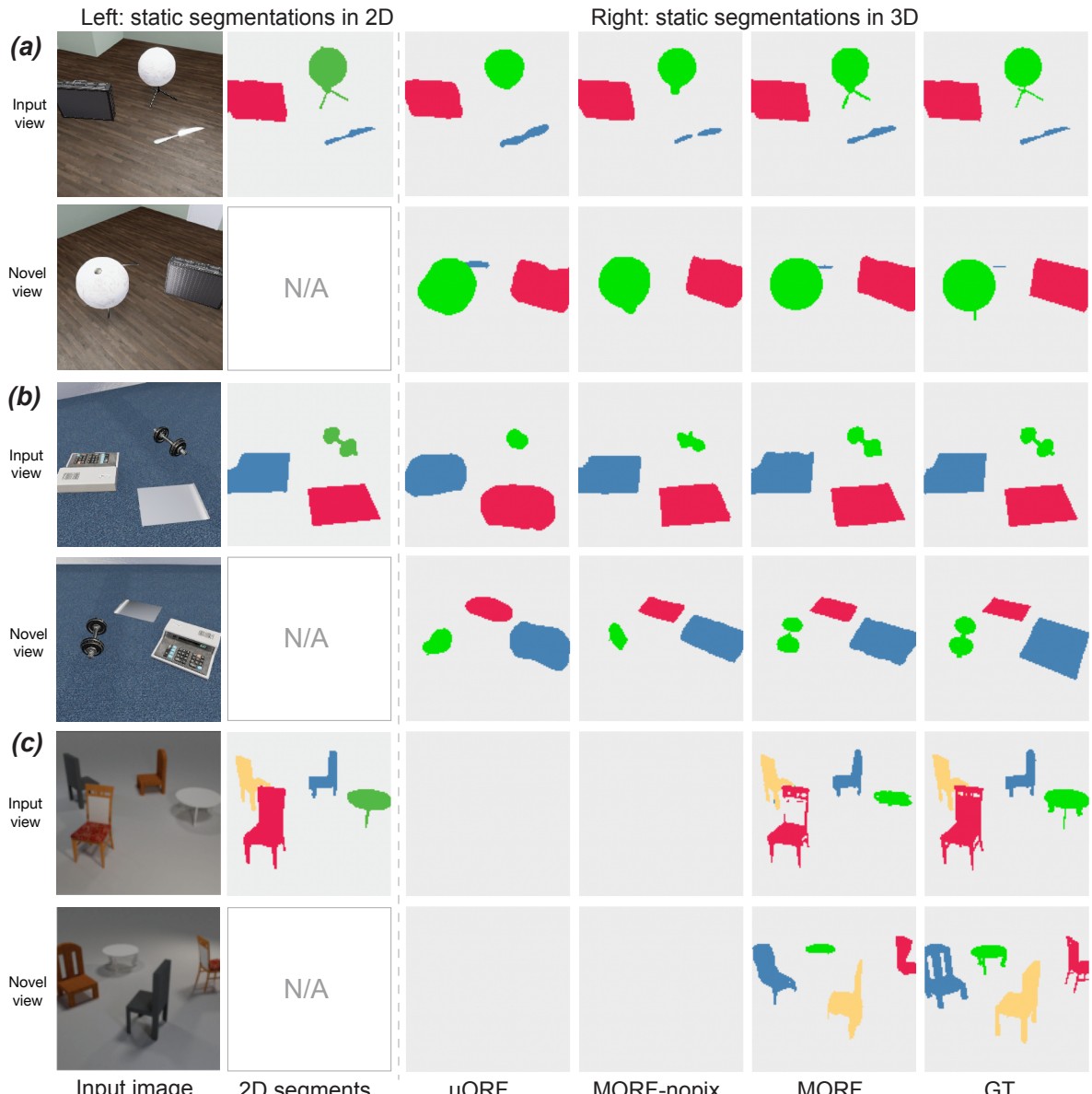

Figure 5: 3D static scene segmentation results on (a) *Playroom*, (b) *Playroom-novel*, and (c) *MultiShapeNet* datasets. The left shows the 2D scene segmentations of the input view predicted by EISEN (Mask2Former for MultiShapeNet), which are used to conditionally infer 3D object radiance fields. The right shows the scene segmentations for both the input view and the novel view. Note that both uORF and MORF-nopix collapse on MultiShapeNet, and thus they cannot generate meaningful object segmentations.

S-CD metric measures the quality of the objects' layout in 3D. Due to the lack of object decomposition, pixelNeRF only outputs a single mesh encompassing both the foreground objects and the background. For a fair comparison of S-CD with MORF, we remove the background meshes by setting the density of the grid points below a z-value threshold to zero before applying marching cubes. We search for the threshold with the best S-CD on a validation set, and use the threshold to calculate pixelNeRF's S-CD.

**Results**. We show the quantitative results in Table 2. MORF outperforms all methods in terms of mIoU, O-CD, and S-CD. Figure 4 shows the quality of mesh reconstructions. uORF (Yu et al., 2022) is only able to recover the coarse geometry of the objects. pixelNeRF (Yu et al., 2020) tends to miss the fine details of small objects and thin objects, while MORF can learn more fine-grained object geometry given a single image.

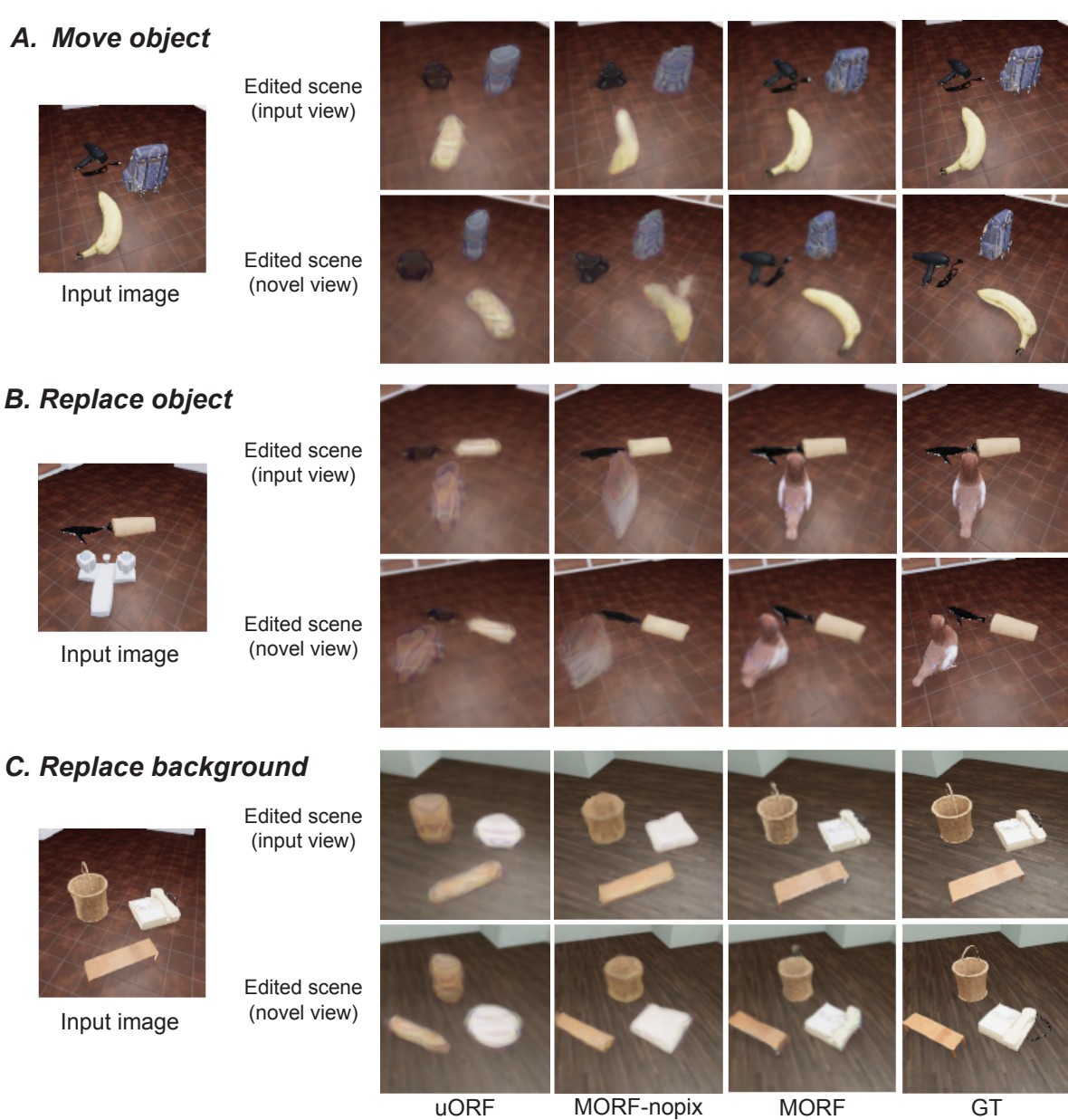

Figure 6: Qualitative results on three scene scene editing tasks. The synthesis of the edited scene from both input and novel views are shown in the top and bottom rows respectively. MORF is able to manipulate individual object radiance fields to generate novel scenes. Both uORF and MORF (without pixel features) are able to perform scene editing to a limited extent, but their reconstructions are blurry. This comparison shows the advantage of using both EISEN segmentations and pixel latents for scene editing tasks.

For pixelNeRF (Yu et al., 2020), O-CD is not reported as it lacks object decomposition. Geometry reconstruction metrics are not reported for SRT, since SRT does not have explicit geometry representations. Figure 5 shows the quality of scene segmentations. Compared to MORF, uORF and the pixel feature-ablated version produce less accurate scene segmentations on *Playroom* and *Playroom-novel*. Furthermore, they both collapse in training and, thus, fail to decompose the scene into multiple objects on *MultiShapeNet*.

| Models | Playroom | | | Playroom-novel | | |
|---|---|---|---|---|---|---|
| | mIoU↑ | O-CD ↓ | S-CD ↓ | mIoU↑ | O-CD ↓ | S-CD↓ |
| uORF | 59.1 | 0.324 | 0.113 | 58.4 | 0.324 | 0.150 |
| pixelNeRF | - | - | 0.133 | - | - | 0.128 |
| MORF-nopix | 65.4 | 0.239 | 0.096 | 61.8 | 0.260 | 0.122 |
| MORF (ours) | **72.7** | **0.208** | **0.078** | **66.4** | **0.224** | **0.110** |

Table 2: Quantitative results on geometry reconstruction

MORF outputs less accurate scene segmentations on *MultiShapeNet* because the 2D segmentations from the pretrained Mask2Former model are prone to generalization error.

### 4.3 Scene editing

We consider three scene editing tasks: moving objects, replacing objects, and replacing the background. For moving and replacing objects, we follow the protocol of uORF (Yu et al., 2022) and select the object that has the largest IoU with the ground-truth masks of the target object for editing.

| Models | Move object | | | Replace object | | | Replace background | | |
|---|---|---|---|---|---|---|---|---|---|
| | LPIPS↓ | SSIM↑ | PSNR↑ | LPIPS↓ | SSIM↑ | PSNR↑ | LPIPS↓ | SSIM↑ | PSNR↑ |
| uORF | 0.381 | 0.573 | 20.7 | 0.384 | 0.575 | 20.8 | 0.371 | 0.593 | 20.4 |
| MORF-nopix | 0.302 | 0.706 | 21.9 | 0.328 | 0.701 | 21.4 | 0.319 | 0.700 | 20.9 |
| MORF (ours) | **0.223** | **0.758** | **22.7** | **0.250** | **0.751** | **22.1** | **0.232** | **0.758** | **21.7** |

Table 3: Quantitative comparison on scene editing tasks

pixelNeRF (Yu et al., 2020) is not comparable on scene editing due to the lack of object decomposition. We report LPIPS, SSIM, and PSNR on the *Playroom-edit* dataset. Ground-truth masks are only used to select which object to edit. EISEN segmentation masks from a pretrained model are used in the feedforward pass of MORF on all the images. We show the results in Table 3 and Figure 6. MORF outperforms uORF on all metrics across the three editing tasks. MORF trained without pixel features results in blurry reconstruction on some of the objects.

### 4.4 Ablation studies

MORF's performance on *Playroom* drops slightly (except for O-CD) when object latents are not used. In contrast, pixel latents are crucial for MORF to perform well on both novel view synthesis and geometry reconstruction. This implies that although object-centricity provides a strong optimization constraint for learning object radiance fields, local pixel features are still critical for modeling the fine details of object textures and shapes.

| Mask types | slot feat. | pixel feat. | Playroom | | | | |
|---|---|---|---|---|---|---|---|
| | | | LPIPS ↓ | SSIM↑ | PSNR↑ | O-CD ↓ | S-CD ↓ |
| EISEN | ✓ | ✗ | 0.266 | 0.727 | 22.8 | 0.239 | 0.096 |
| EISEN | ✗ | ✓ | 0.170 | 0.776 | 24.3 | 0.198 | 0.079 |
| EISEN | ✓ | ✓ | 0.162 | 0.781 | 24.4 | 0.208 | 0.080 |
| GT | ✓ | ✓ | 0.140 | 0.791 | 25.0 | 0.147 | 0.036 |

Table 4: Ablation study results.

Finally, we compare MORF trained with EISEN segmentation masks with MORF trained with GT masks. Unsurprisingly, the latter shows better performance in all metrics. Qualitatively, we observe that MORF occasionally misses small objects or parts, which is likely a direct consequence of EISEN failing to accurately segment these fine-scale scene elements. This indicates that improving unsupervised 2D segmentation should lead to further improvement of MORF's 3D representation learning.

## 5 Conclusion

In this work, we propose Movable Object Radiance Fields (MORF), which scales unsupervised 3D object-centric scene representation learning to complex and diverse multi-object scenes. Our key technical innovation is to allow learning 3D representations by integrating and lifting a 2D object segmentation module. We show that once high-quality segmentations are obtained through an independent unsupervised module, we can learn 3D object-centric representations in complex category-agnostic scenes. This innovation addresses the limitations of the previous methods and allows substantial progress to be made on this difficult and important problem.

We demonstrate that MORF enables faithful photometric and geometric reconstruction of scenes with unseen configurations from a single view, generalizes well to unseen object categories, and supports complex editing tasks. We believe that our positive results suggest the promise of further scaling unsupervised 3D factorized representation learning to more complex scenes.

## Acknowledgments

This work is in part supported by NSF RI #2211258 and #2338203, ONR MURI N00014-22-1-2740, ONR N00014-23-1-2355, the Stanford Institute for Human-Centered AI (HAI), Google, and Samsung.

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

## A  Appendix

### A.1  Dataset details

The Playroom dataset is generated with ThreeDWorld  (Gan et al., 2020) using custom code. The dataset contains 15,000 training scenes and 600 test scenes. Each scene includes 3 objects randomly sampled from a set of unique 2,086 object models in 231 object categories. The objects have complex geometry and textures. Table 5 and Figure 7 show the comparison between the Playroom dataset and the MultiShapeNet-Easy dataset. We also generate an additional dataset *Playroom-diverse* containing 30 unique background environments, 10 times more than the *Playroom* dataset.

We simulate 2 video frames for each scene, with a randomly selected object invisibly pushed at the first frame to generate object motion. At each time step, we render 4 views at resolution 128 with random camera azimuth angles. It is important to note that we only use the video frame pairs for training EISEN. Once EISEN is trained, only a single static image is required for inference. We only use static images at a single time step for learning radiance fields. Figure 8 shows examples of the video frame pairs from the dataset, along with the EISEN segmentations and ground truth segmentation masks.

**Playroom**

**Playroom-diverse**

**MultiShapeNet - Easy**

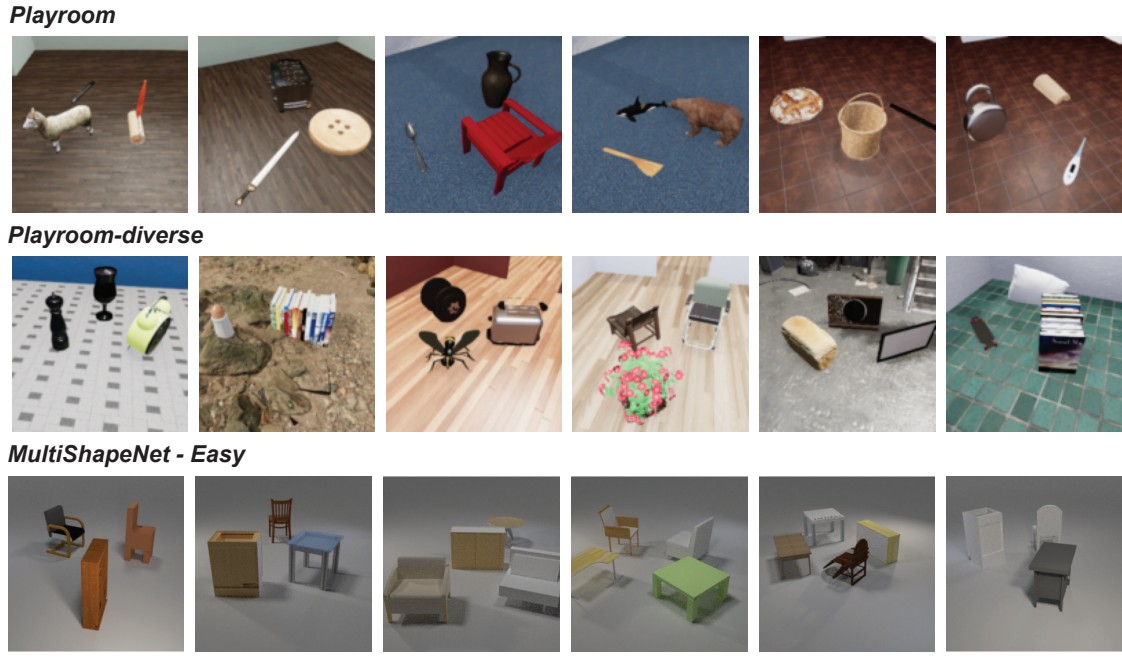

Figure 7: Qualitative comparison between Playroom and MultiShapeNet.

| Dataset | scenes | objects | categories | environment maps | object per scene | views |
|---|---|---|---|---|---|---|
| Playroom | 15,000 | 2,000 | 231 | 3 | 3 | 4 |
| Playroom-diverse | 30,000 | 2,000 | 231 | 30 | 3 | 4 |
| MultiShapeNet-Easy | 80,000 | 12,000 | 3 | 1 | 2-4 | 3 |

Table 5: Dataset statistics comparison.

### A.2  Implementation details

**Convolutional backbone encoder.**  To compute object and pixel latents, we use two separate convolutional encoders to extracts feature pyramids from the input image respectively. We show the encoder

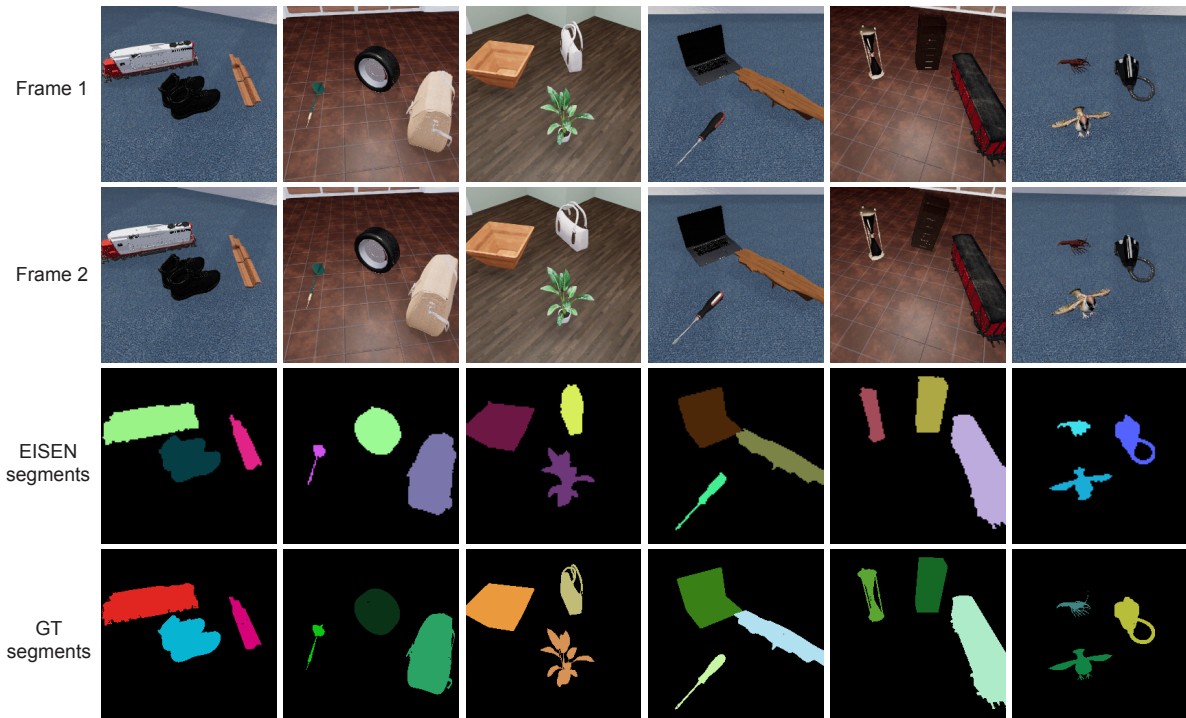

Figure 8: Playroom video frames, EISEN segmentation, and ground truth segmentation.

architectures in Table 6 and Table 7. All the encoder weights from initialized from scratch without any pretraining. All the activation layers are ReLU.

**NeRF decoder.** We adapt the NeRF architecture from uORF (Yu et al., 2022). We show the architecture detail in Figure 9. The input to the NeRF network consists of positional embeddings, ray directions, object latents, and pixel latents. In all experiments, we set the positional embedding frequency to 5. Note that we do not apply high-frequency encoding to the ray directions. The feature dimension of object and pixel latents are set to 64 respectively. Following uORF, the background decoder is slightly different from the foregorund decoders in that it does not have the second last layer and third last layer.

| Layer name | Input shape | Output shape | Concatenate |
|---|---|---|---|
| conv1 | 64×64×7 | 64×64×64 | |
| conv2 | 64×64×64 | 32×32×64 | |
| conv3 | 32×32×64 | 16×16×64 | |
| conv4 | 16×16×64 | 16×16×64 | |
| Upsample | 16×16×64 | 32×32×64 | |
| conv5 | 32×32×128 | 32×32×64 | with conv2 |
| Upsample | 32×32×64 | 64×64×64 | |
| conv6 | 64×64×128 | 64×64×64 | with conv1 |

Table 6: Backbone encoder for computing object latents.

## A.3 Experiment details

**Training.** In each batch, we input one image and render 2 reconstructed images for supervision, one from the input view and the other from a randomly selected novel view. We render each pixel with 128 samples. For perceptual loss, we uses the output of the fourth convolutional block in a VGG16 (Simonyan & Zisserman, 2014) pretrained on ImageNet. We use Adam optimizer with an initial learning rate 0.0003. We adopt a

| Layer name | Input shape | Output shape | Concatenate |
|---|---|---|---|
| block1 | 64×64×3 | 32×32×64 | |
| block2 | 32×32×64 | 16×16×64 | |
| block3 | 16×16×64 | 8×8×128 | |
| block4 | 8×8×128 | 4×4×256 | |
| Upsample | 4×4×256 | 8×8×256 | with block3 |
| Upsample | 8×8×384 | 16×16×384 | with block3 |
| Upsample | 16×16×448 | 32×32×448 | with block2 |
| Upsample | 32×32×448 | 64×64×448 | with block1 |
| Linear | 64×64×512 | 64×64×64 | |

Table 7: Backbone encoder for computing pixel latents.

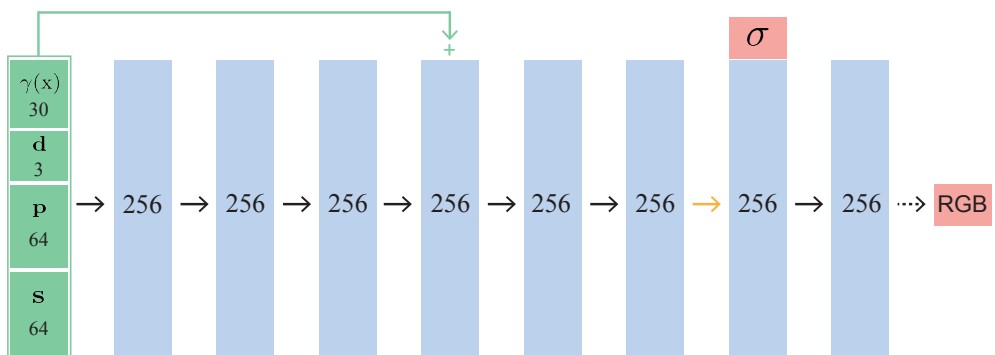

Figure 9: NeRF decoder network architecture. Input vector are shown in green, full-connected hidden layers are shown in blue, output vectors are shown in red. The input vector is a concatenation of positional embeddings $\gamma(\mathbf{x})$, ray direction $\mathbf{d}$, pixel latents $\mathbf{p}$, and object latents $\mathbf{s}$. The number inside each block indicates the dimension of the vectors. Black arrows indicate layers with ReLU activations, orange arrows indicate layers with no activation, dashed black arrows indicate layers with sigmoid activation. "+" indicates vector concatenation.

linear learning rate warm-up for the first 1000 iterations. The learning is exponentially decreased by half for every 200 thousand iterations. We train our model with a batch size of 16 for 500 thousand steps, which takes about 10 days on 8 NVIDIA A40 GPUs.

**Inference** Figure 10 shows the inference procedure. MORF takes a single view of a new scene as input and infers object and background radiance fields in a single forward pass. Given an arbitrary camera pose, a novel view can be synthesized via compositional rendering.

**Spatial sampling for memory efficiency.** Rendering multiple radiance fields for all the objects in a scene is memory expensive, especially when there are a large number of objects per scene. To improve the memory efficiency of the training, we only compute object radiance field at a small subset of world coordinates. Specifically, we only compute object radiance field at world coordinates whose projected 2D coordinates onto the input view resides in the segmentation mask of the corresponding object. Otherwise, we set the density as zero. This significantly reduces for the number of forward passes through the NeRF decoder networks. We add padding around the object segments to account for potential incomplete segmentation due to occlusion.

**Metric details.** We use PSNR and SSIM from the Pytorch Image Quality (PIQ) package as in uORF. LPIPS is computed using the code provided by the LPIPS authors Zhang et al. (2018) after normalizing the pixel values to [-1, 1]. We compute Scene Chamfer Distance (S-CD) on the foreground object meshes in a scene. Points are uniformly sampled on the mesh surface to create a dense point cloud; then $N$ points are randomly sampled from the point cloud, where $N$ is 1,024 per object mesh and 3,072 per scene mesh. We normalize the point cloud coordinates into a unit cube before CD calculation.

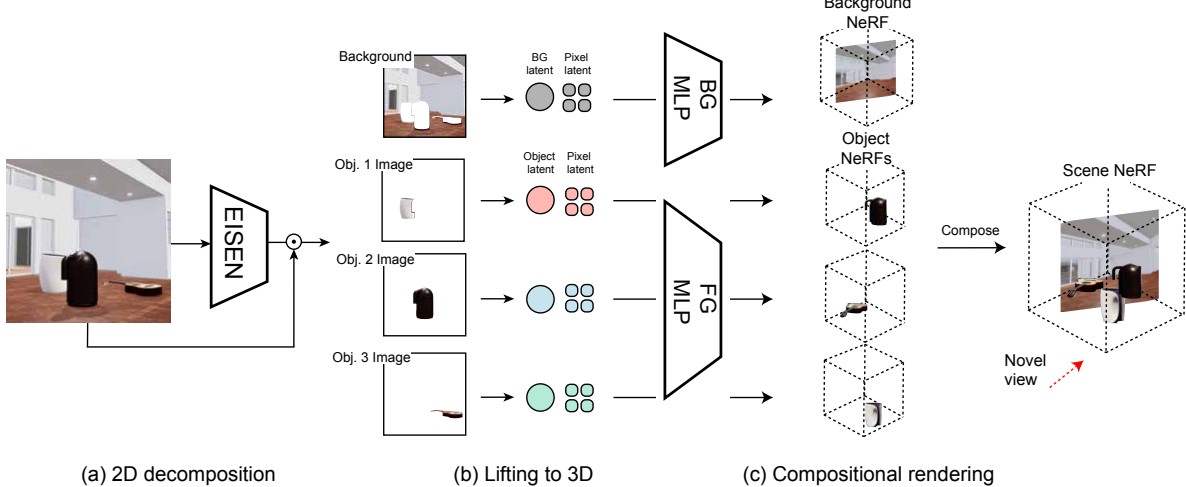

(a) 2D decomposition  (b) Lifting to 3D  (c) Compositional rendering

Figure 10: **Model inference**. MORF takes as input a single image of a scene with potentially diverse objects, and infers 3D object and background radiance fields. (a) MORF infers a set of object masks using a pre-trained segmenter. Masked images are used to generate object radiance fields. (b) MORF generates object radiance fields conditioned on the latent object and pixel codes. (c) MORF reconstructs the novel view via compositional rendering.

| Models | Playroom | | Playroom-novel | |
|---|---|---|---|---|
| | ARI↑ | mIoU ↑ | ARI ↑ | mIoU ↑ |
| SlotAttn | 7.7 | 22.8 | 7.5 | 22.5 |
| Unsup. SAVi | 2.0 | 14.6 | 2.0 | 14.8 |
| DOM | 73.6 | 57.2 | 71.9 | 56.4 |
| EISEN | **93.3** | **87.3** | **92.2** | **87.4** |

Table 8: Quantitative comparison of scene segmentations in 2D.

## A.4  Static scene segmentation in 2D

We evaluate the segmentation quality of EISEN and compare it to other unsupervised segmentation methods: Slot attention  (Locatello et al., 2020), unsupervised SAVi  (Kipf et al., 2021), and DOM  (Bao et al., 2022). To evaluate segmentation quality, we compute both the Adjusted Rand Index (ARI) and mean Intersection over Union (mIoU) metrics, following previous works in unsupervised 2D segmentations. As shown in Table 8 and Figure 11, Slot attention and unsupervised SAVi remain struggling with segmenting the Playroom objects. DOM uses optical flow to self-supervise the segmentation masks. However, it underperforms EISEN and outputs incomplete object segmentations, especially for objects with complex geometry details. The output segmentations of the baseline methods are not accurate enough to be used for training downstream object radiance fields.

## A.5  Additional qualitative results

**More novel view synthesis results**  Figure 12 shows additional view synthesis results on *Playroom* dataset. Figure 13 shows additional results on *Playroom-diverse* dataset. Figure 14 shows additional results on *MultiShapeNet* dataset. For each image, the top row shows the reconstructions from the input view, the bottle row shows the reconstructions from a randomly selected novel view.

**Scenes with different numbers of objects**  We create additional testing scenes with a random number of objects between 1 and 6. We show MORF is able to generalize well to test scenes with a different number of objects. Figure 15 shows qualitative results of the challenging scenarios with 4 to 6 objects in a scene.

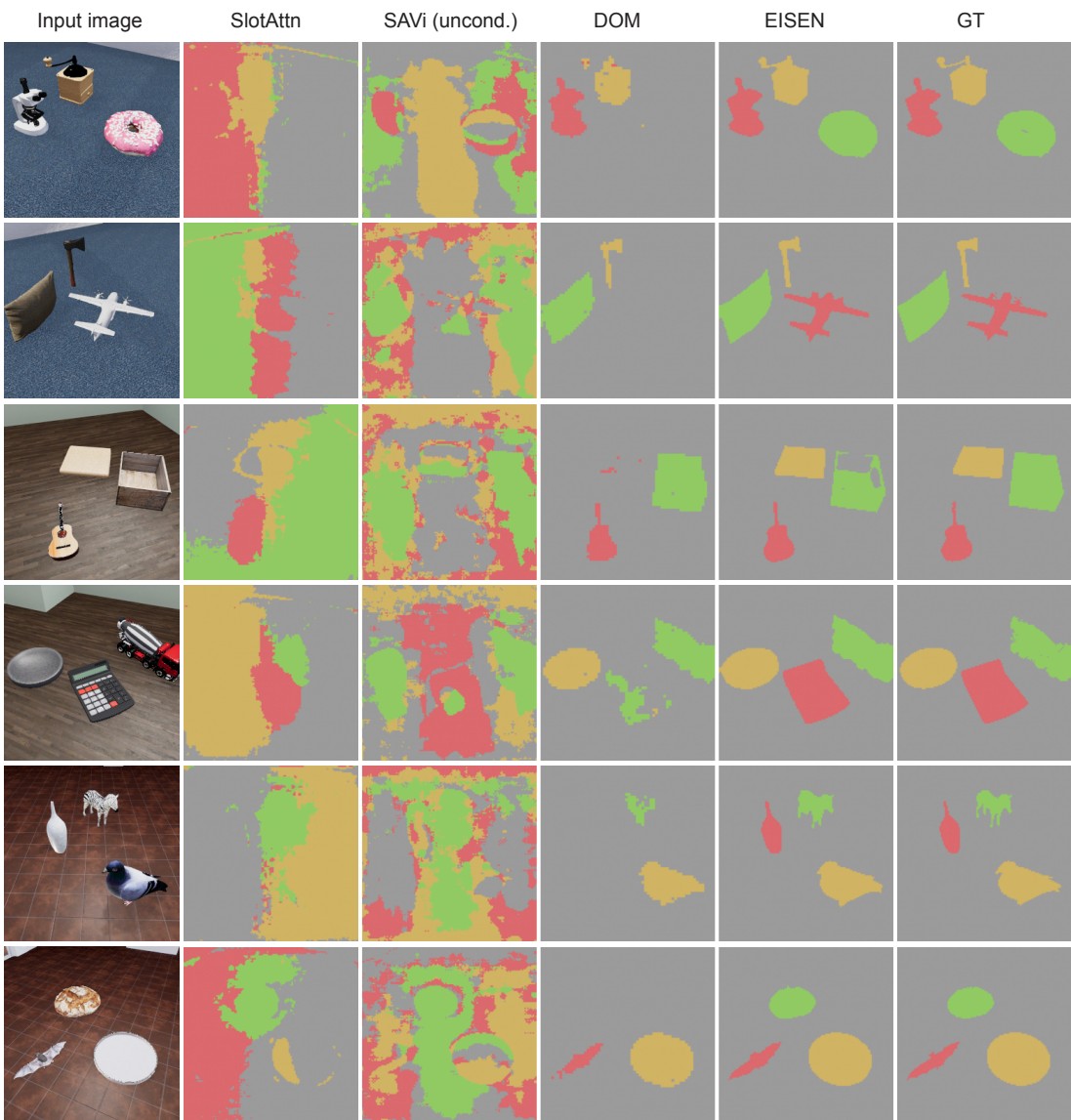

Figure 11: Qualitative results on static scene segmentation in 2D.

**Scenes with object occlusion**  The training and validation sets contain examples with partial object occlusion since we are rotating the camera with random azimuth angles. We show qualitative examples of partial object occlusion in the first two rows of Figure 16. To further test the ability of MORF in handling scenes with more object occlusion, we create additional test scenes with objects placed closer to each other, as shown in the last two rows of Figure 16. MORF is generally capable of handing partial object occlusion and performing approximate amodal completion of the occluded objects. The quality of geometric reconstruction slightly deteriorates as compared to the unoccluded objects in the input view.

**Scenes with object stacking**  *Playroom* dataset simulates an infant playing environment with toys placed on the floor. In real-world scenarios, objects are often stacked on top of one another. We create additional scenes with three types of stacking configuration: (a) a cup on top of a coaster, (b) a food item on the plate, and (c) an object on top of a microwave. We generate 1000 scenes for each stacking configuration and fine-tune a MORF model on the additional scenes. Figure 17 shows the qualitative results of novel view synthesis on held-out scenes.

| Models | LPIPS↓ | SSIM↑ | PSNR↑ |
|---|---|---|---|
| uORF | 0.336 | 0.577 | 19.8 |
| MORF-nopix | 0.323 | 0.580 | 19.9 |
| MORF (ours) | **0.064** | **0.903** | **31.0** |

Table 9: Quantitative comparison on real-world scenes

**Generalization to real world scenes**  We go beyond the synthetic data to test the pretrained model's generalization on real images. Given a few real photos taken using a cellphone, we first use an unsupervisedly pretrained EISEN to compute the segmentations, followed by novel views reconstruction using MORF pretrained on *Playroom*. We show the reconstructions and 3D segmentations in Figure 18. Our method is able to predict plausible geometry and segmentations for all the objects from a novel view.

**Training on real-world scenes**  Besides training on synthetic data, we train on a real-world dataset including scenes with four plant pots or vases on a tabletop decorated with tablecloths. We gathered 9 plant pots, 8 vases, and 6 tablecloths and captured 745 scenes for training and 140 for evaluation, each with 3 images from different poses. We cannot train EISEN on this dataset since there is no motion. We thus replace EISEN with a Mask2Former (Cheng et al., 2021) model pretrained on the COCO dataset to compute the 2D segmentations of the dataset. Table 9 and Figure 19 show novel view synthesis results on the testing real-world scenes. MORF synthesizes the novel view accurately, whereas uORF fails to decompose the scenes, resulting in inaccurate 3D object radiance fields.

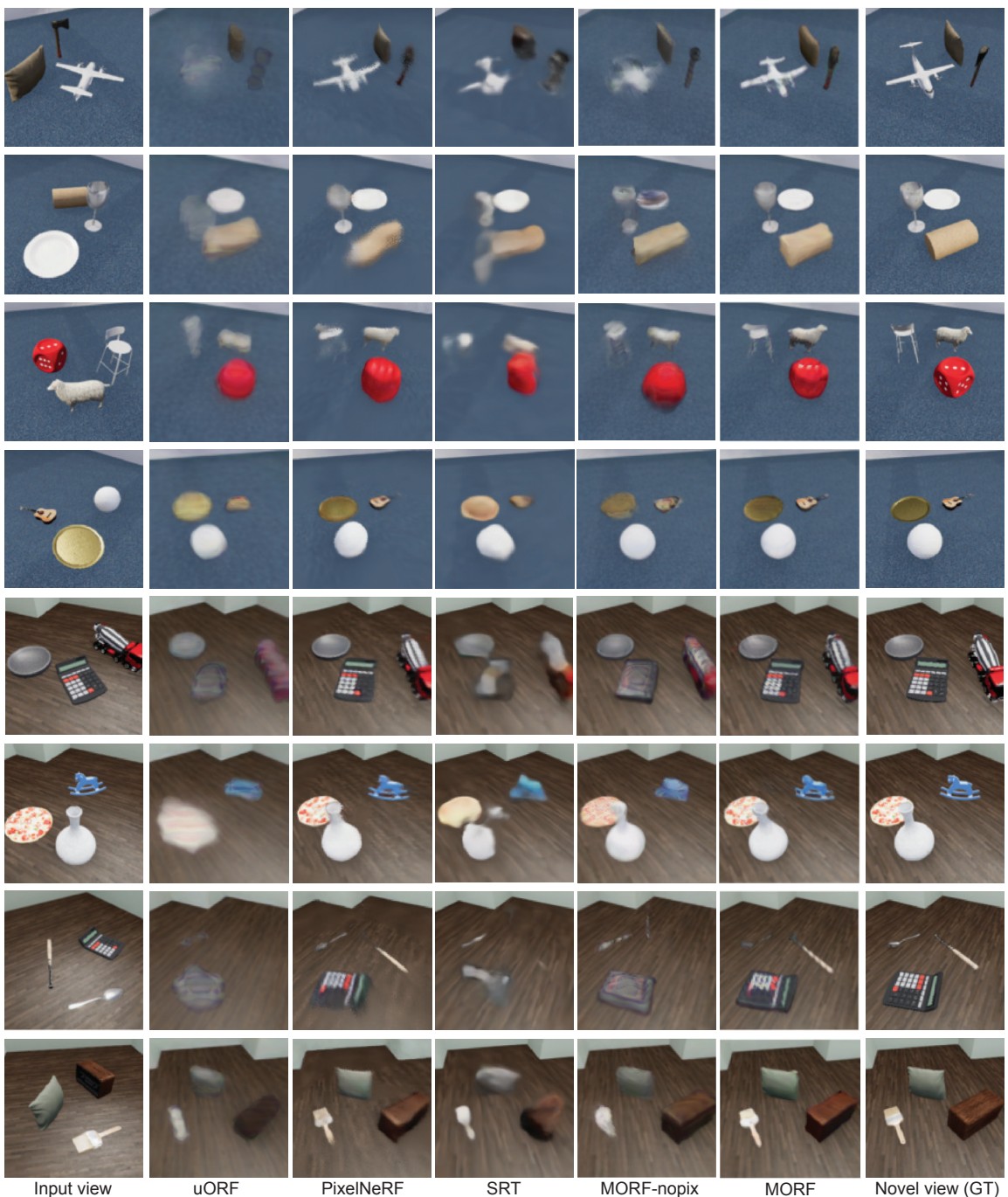

Figure 12: Additional novel view synthesis results on *Playroom*

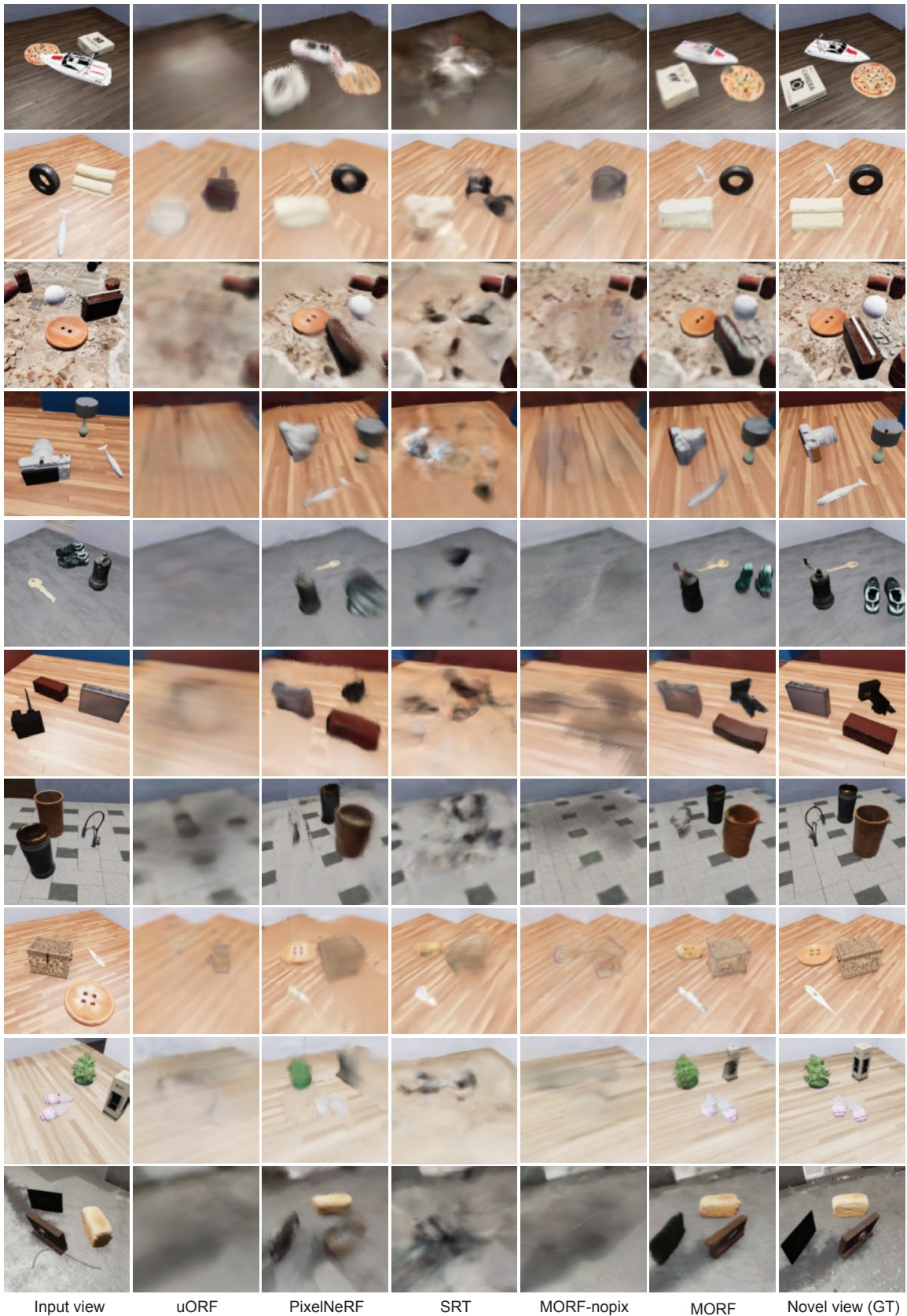

Input view     uORF     PixelNeRF     SRT     MORF-nopix     MORF     Novel view (GT)

Figure 13: Additional novel view synthesis results on *Playroom-diverse*

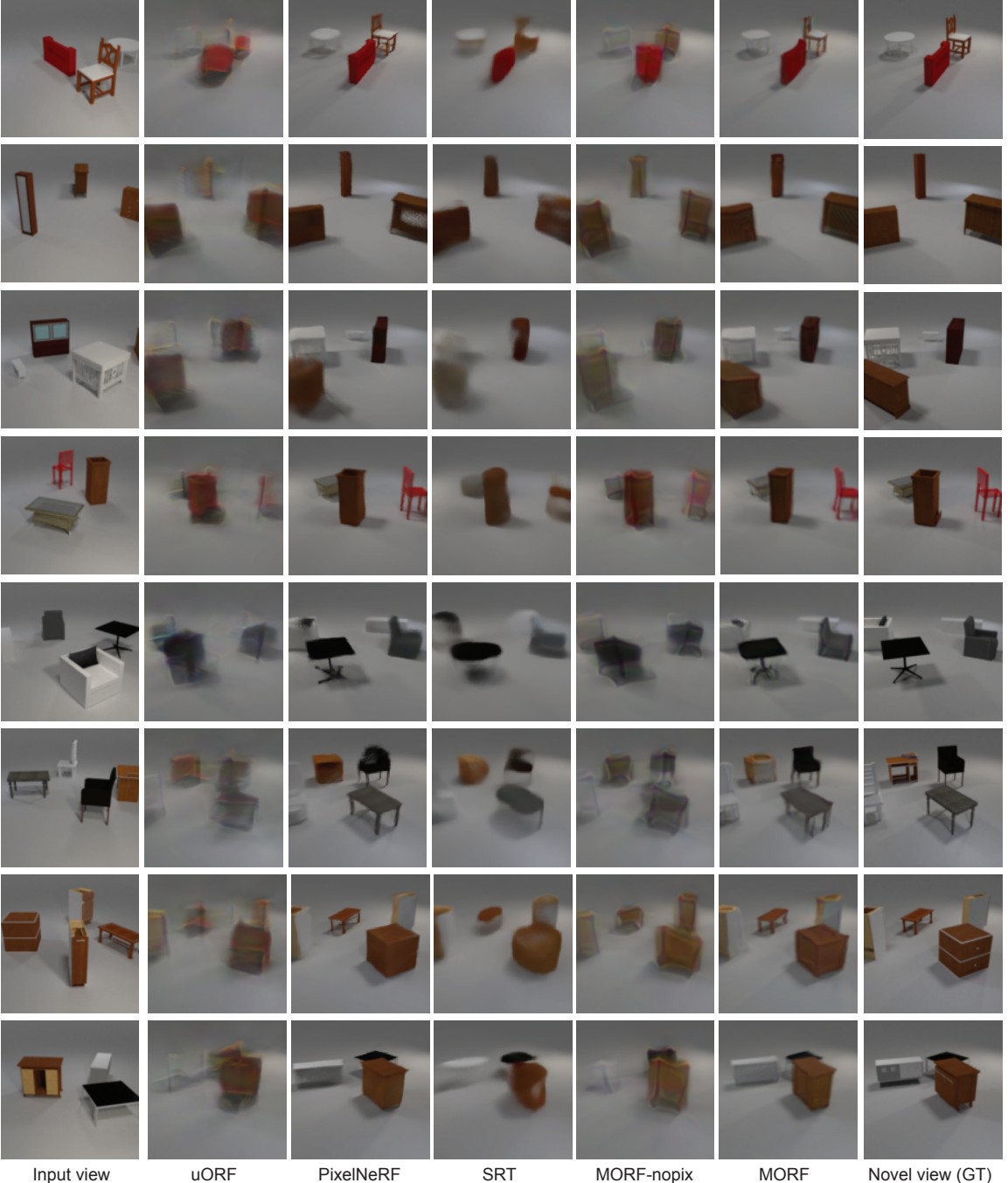

Figure 14: Additional novel view synthesis results on *Playroom*

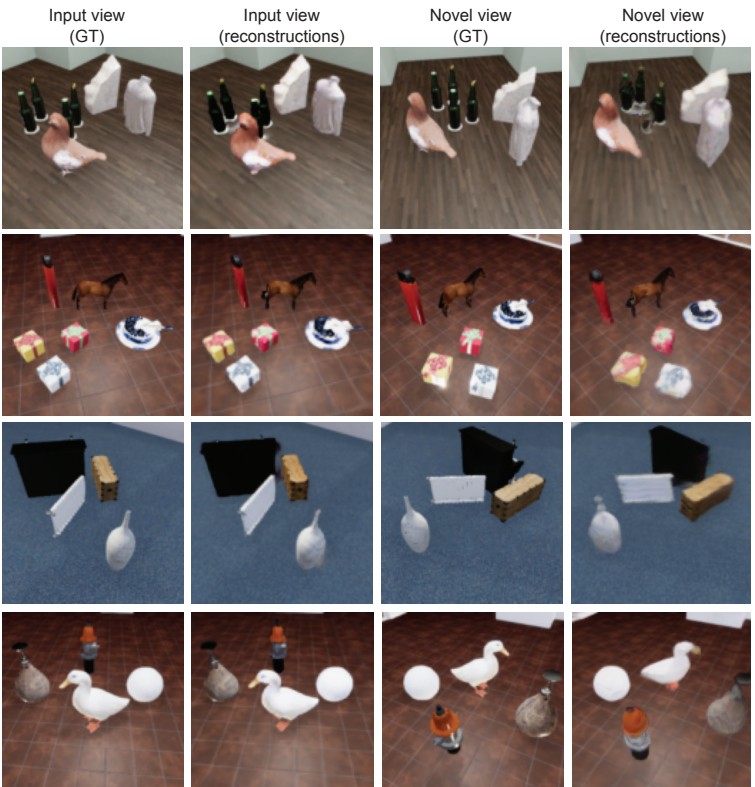

Figure 15: Novel view synthesis results of testing scenes with arbitrary objects

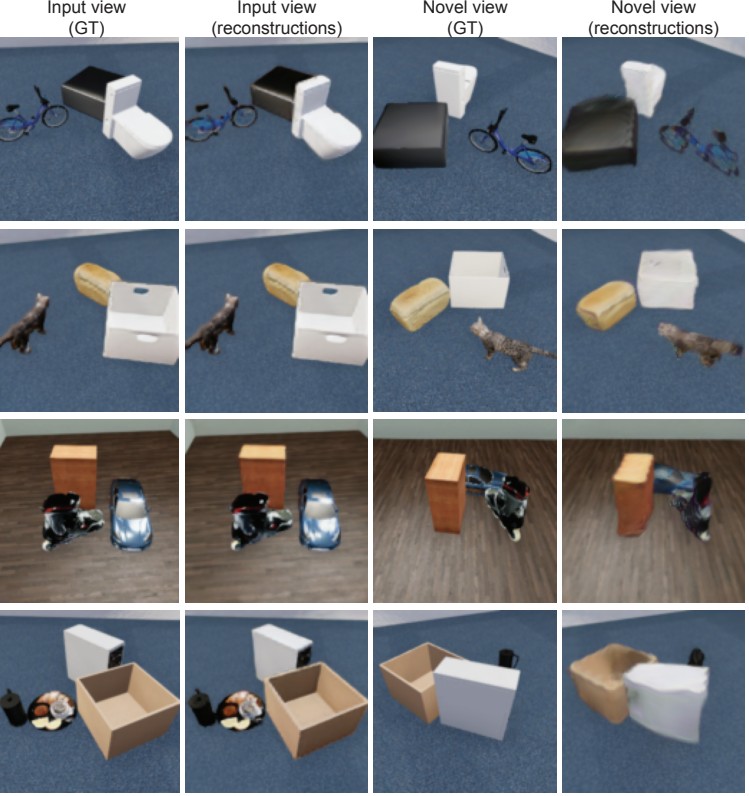

Figure 16: Novel view synthesis results of scenes with object occlusion

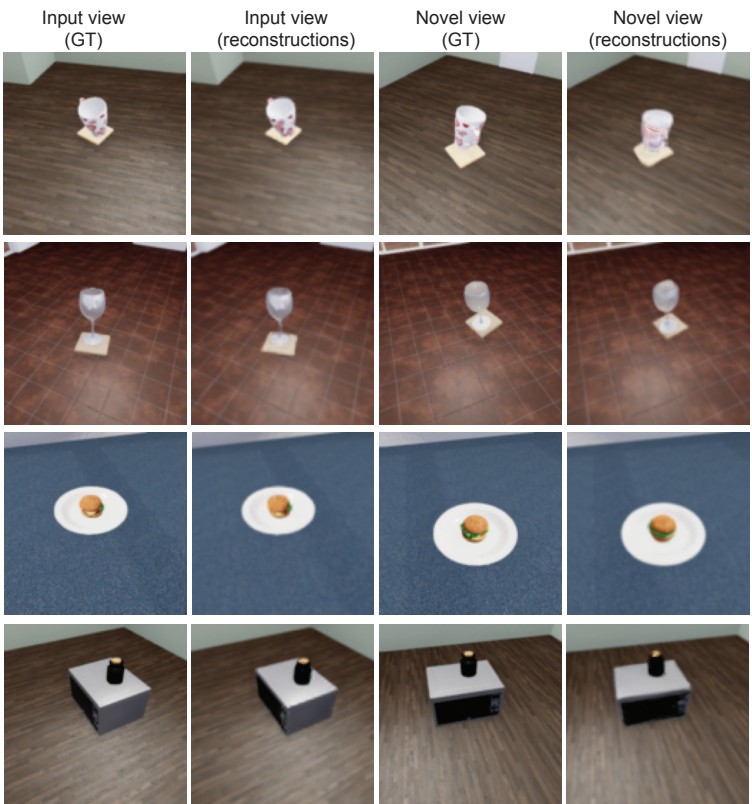

Figure 17: Novel view synthesis results of scenes with stacking configuration

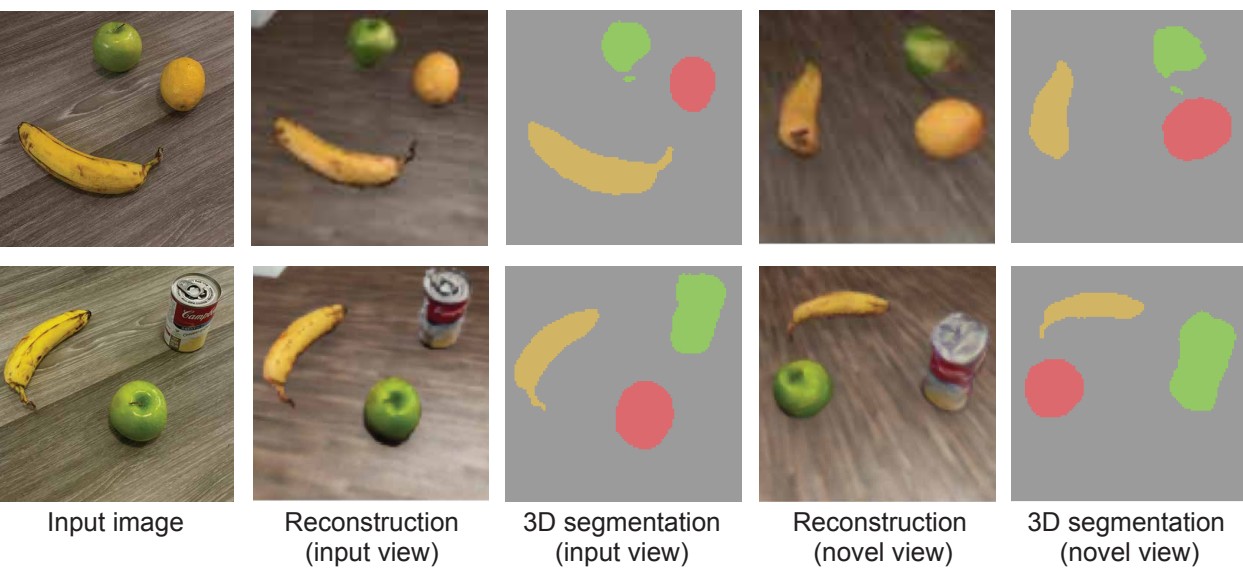

Figure 18: Novel view synthesis and 3D scene segmentations on real images taken using a cellphone.

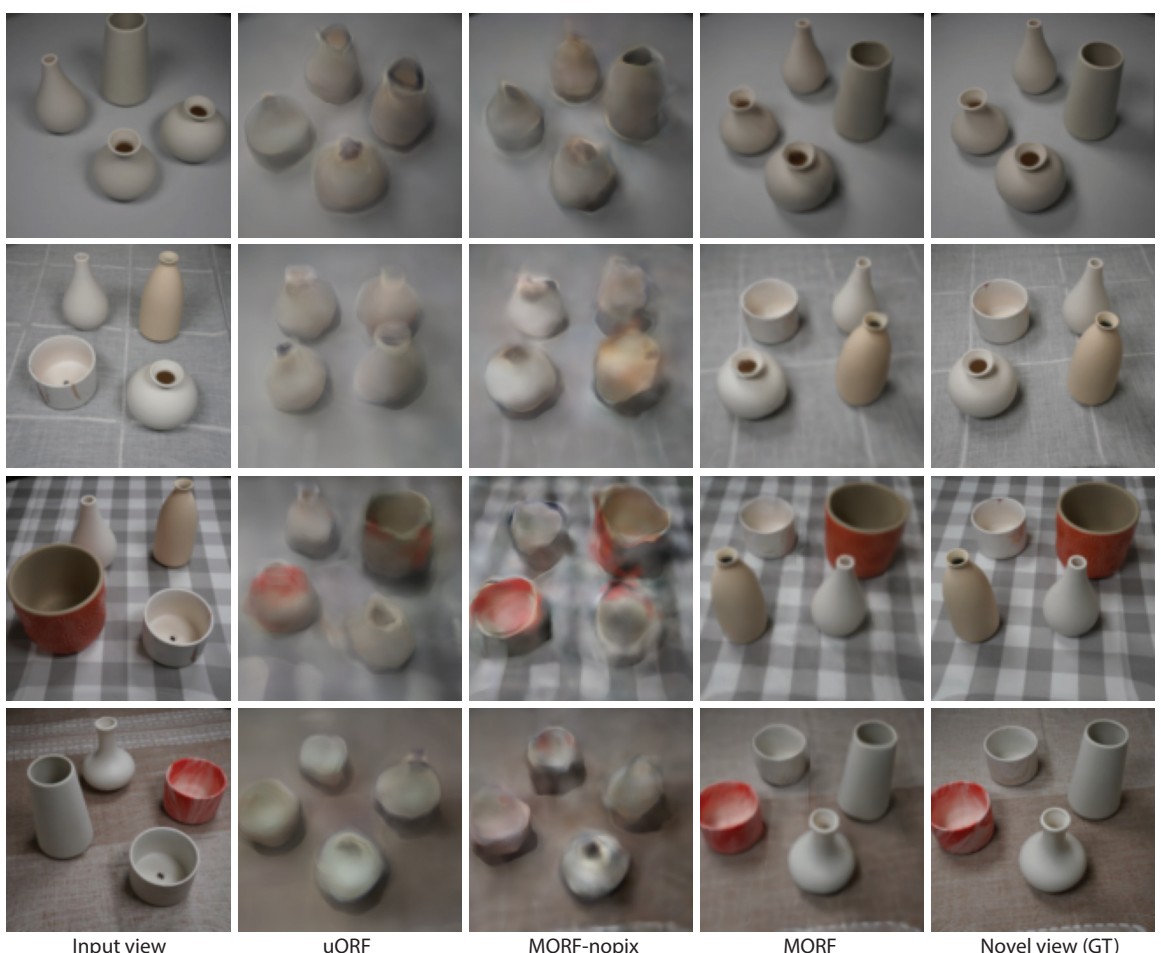

Figure 19: Novel view synthesis results on real-world scenes

