# OpenReview forum: "Unsupervised 3D Scene Representation Learning via Movable Object Inference"
_TMLR — Accepted by TMLR_

### Review · Reviewer_BY36 · 2023-10-01

**Summary Of Contributions:**

The paper proposed a new unsupervised object-centric 3D scene representation learning framework called MORF. Leveraging 2D unsupervised segmentation approaches, the algorithm disentangles the foreground/background images, and learns different representations for different object instances. The approach is able to provide a wide varieties of the tasks, such as 3D reconstruction, image editing, etc. Experimental results on 3D datasets suggest that MORF has potentials of reconstructing higher quality 3D geometries and have improved scene editing capabilities against other algorithms.

**Audience:**

Yes

**Claims And Evidence:**

Yes

**Requested Changes:**

There seems to be some poorly formatted documents/texts starting from Sec. 4.3, it's a good idea to fix it for future revisions. The following sections of the paper looks like to be completed in a rush.

**Strengths And Weaknesses:**

[Strengths]
- The paper is well-written, and I think they are relatively easy to follow. I think reproducibility should be ok as well
- The algorithm is able to provide wider ranges of tasks, such as generalizing to more classes of objects, etc. Also, experimental results are slightly sharper and higher quality compared with other approaches.
- The claims in this paper are relatively grounded, and the qualitative results demonstrated that the proposed algorithm is doing well.

[Weaknesses]
- The paper writing is generally OK, but I don't fully get the motivation of using an unsupervised optical flow approach (EISEN). I know it's one of the state-of-the-art algorithms for such tasks, but utilizing more training data (pairs of images) may grant MORF an unfair advantage over other approaches. It's a little bit challenging to justify the novelties of the approach.
- The EISEN algorithm is pre-trained using Sintel synthetic datasets, therefore I wonder if the generalization ability of the approach could potentially be better if we pre-train on real-world optical flow datasets.

- nit: What's authors intuition of using multiple FG MLPs for each objects? Will that further reduce the artifacts in the results?

---

> ### Author Response · Authors · 2023-12-17
> **Response to Reviewer BY36**
>
> Thank you for your constructive comments and feedback!  We have revised the paper based on your feedback, and respond to the individual points below.
>
> 1. *I don't fully get the motivation of using an unsupervised optical flow approach (EISEN)*
>
>     We clarify that the goal of this paper is to solve the important problem of unsupervised object-centric 3D representation learning from a single image of complex scenes, which remains a challenging and unsolved problem. Existing methods are not scalable to datasets with complex object textures, resulting in inaccurate 2D segmentations and 3D radiance field representations. Therefore, we propose lifting unsupervised 2D object segmentation to 3D, and EISEN was the state-of-the-art unsupervised object segmentation approach at the time of submission, as shown by the comparison study in supplementary A.4.
>
> 2. *utilizing more training data (pairs of images) may grant MORF an unfair advantage over other approaches*
>
>     We note that a fundamental bottleneck of existing unsupervised 3D object discovery methods is that they cannot discover complex objects due to the lack of object priors. We inject this prior by integrating a pretrained EISEN (an unsupervised 2D object segmentation method). Therefore, our contribution is to allow leveraging unsupervisedly learned 2D object prior to improving 3D object discovery. All other approaches cannot leverage such important object priors.
>
>     We also note that, since EISEN is a pretrained module, MORF only requires static multi-view images for training, which is the same data requirement as other approaches.
>
> 3. *It's a little bit challenging to justify the novelties of the approach.*
>
>     The key conceptual innovation of our method is to leverage unsupervisedly learned 2D object priors to improve 3D object-centric representation learning. We show that once high-quality 2D object segmentations are obtained through an independent unsupervised method, we can learn 3D object-centric representations in complex category-agnostic scenes. This innovation addresses the limitations of the previous methods and allows substantial progress to be made on this difficult and important problem. We show that our proposed approach achieves much better performances than the SOTA object-centric method uORF. We also show that object-centricity is important for accurate single-image novel view synthesis and leads to stronger performance as compared to non-object-centric methods SRT and PixelNeRF.
>
> 4. *The EISEN algorithm is pre-trained using Sintel synthetic datasets, therefore I wonder if the generalization ability of the approach could potentially be better if we pre-train on real-world optical flow datasets.*
>
>     We clarify the EISEN algorithm is not pre-trained using Sintel synthetic datasets. EISEN is pretrained on real videos using optical flow as self-supervision signals to learn 2D object segmentations, and the optical flow is computed using the RAFT algorithm [1]. RAFT is pre-trained on Sintel datasets. It might help by pretraining RAFT on real optical flow datasets, yet we consider pretraining RAFT out of our scope.
>
> 4. *What's authors intuition of using multiple FG MLPs for each objects? Will that further reduce the artifacts in the results?*
>
>     We clarify that our approach effectively only uses a single FG MLP. In particular, we represent each object by the same FG MLP and an object latent. This allows learning a generalizable object representation.
>
> 5. *Formatting issue*
>
>     We have updated the draft and fixed the error.
>
> Reference:
>
> [1] RAFT: Recurrent All-Pairs Field Transforms for Optical Flow, Zachary Teed, Jia Deng

---

### Review · Reviewer_qD6L · 2023-10-15

**Summary Of Contributions:**

This work presents a new method, MORF, for learning compositional 3D representations of scenes of objects. The proposed approach has the ability to render novel views of scenes, extract the 3D shape of objects, move objects, replace objects, and replace the background. The proposed approach consists of
- unsupervised 2D segmentation (based on the EISEN method, which learns to segment scene entities that appear to be movable)
- global and local latent feature extraction for foreground objects and background
- foreground and background neural fields conditioned on local and global features
- weighted average-based combination of the foreground and background neural field outputs for scene-level rendering


In the first evaluation setting, training is done on synthetic scenes of 3D objects from the Playroom dataset (2000 object instances, 231 categories) in 3 different environments, and evaluation is done on scenes from 100 held out objects. In the second evaluation setting, the model is trained and tested on scenes of 3D objects from MultiShapeNet (12K objects, single simple environment). The proposed approach outperforms prior work at view synthesis quality, 3D object shape reconstruction accuracy, and scene editing tasks.

**Audience:**

Yes

**Broader Impact Concerns:**

Currently, there are no concerns for ethical implications.

**Claims And Evidence:**

No

**Requested Changes:**

My main requests for changes are:
- (Critical) Additional explanation in the draft about what is the reasoning behind the design decisions of how the training/testing data is generated w.r.t the points discussed above? How do these decisions limit what we can conclude about the capability of MORF? This might be better than prior work, but what are the arguments justifying the way the task is implemented in the synthetic domain? Is this complex enough, and if so, why?
- (Between critical and "simply strengthening the work", especially because it would require new data and experiments) One of the strengths of synthetic data is that it allows us to gradually make testing data more challenging across different axes (e.g. increasing the number of objects, increasing the amount of occlusion, using completely novel environments). Adding additional results that take advantage of this would greatly improve the paper, and help us understand how well MORF actually works, and what should be the goals of future work.

If there improvements are added and the results are good, the evidence would be much stronger for the claims in the paper. I am currently selecting that "Claims and evidence are not supported by accurate, convincing and clear" evidence, because while the accuracy and clarity is good, there is large room for improvement (in terms of complexity and realism) of the implementation of the task in the synthetic domain (see weaknesses).

**Strengths And Weaknesses:**

**Strengths**

- MORF is a well designed composition of existing modules that enables state of the art performance. In order to obtain a per-object 3D representation, it makes sense to segment a 2D image into a set of segments based on a learned prior of whether they appear to be movable or not (the EISEN method), and then extracting global and pixel-level features for the segmented objects to condition neural fields.
- High performance relative to prior methods uORF, PixelNerf, SRT, for scene-level representations.

**Weaknesses**

When training and evaluating computer vision systems in the synthetic domain, it's essential to carefully design the data generating process to faithfully represent the key challenges of a real-world perception task. In this paper, the synthetic data is generated in a way that doesn't capture all of these challenges sufficiently, which in turn limits what we can conclude about the utility of this representation for downstream tasks (e.g. manipulation) and the limitations of the proposed approach.

For compositional 3D representations of scenes of objects, the main challenges are:
- In terms of encoding semantic properties
     - generalization to novel environments (different surfaces the objects are placed on) and lighting
     - generalization to novel object instances of categories seen during training
     - generalization to instances of novel object categories not seen during training
- In terms of encoding geometric properties
     - handling an arbitrary number of objects in a scene
     - handling camera pose variability
     - accurately capturing object scales, locations and rotations in 3D - in the real world, objects can have complex spatial relationships like being close and far from each other, as well as being put inside or on top of other objects.

The current data generation setup is either limited w.r.t. a majority of these challenges, or the details to determine this are missing:

- in Playroom there are only 3 environments and they're used both for training and testing. MultiShapeNet has one environment.
- in Playroom, there are 100 novel objects, but it's not clear whether these are from novel categories or not, or how different they are from the objects used in the training set. In MultiShapeNet it's not clear whether the training/validation objects are distinct.
- there are a most 4 objects in each scene
- the objects appear to be well-spaced, and the camera is at a fixed height. As a result, it appears that object-object occlusion doesn't happen very often, which is unrealistic.
- objects have similar scales and are placed in canonical poses. For example in the real world, an apple is much smaller than a table or a bowl and is usually found on top of a table or inside a bowl.
- the camera is at a fixed height, greatly simplifying the scene-level geometric relationships that need to be learned.


Building up to a synthetic training and evaluation setup that faithfully captures all of these will be a gradual process, but given the nature of the task and the capabilities of current 3D graphics systems and freely available 3D assets, I believe the proposed data generation system should be more complex and diverse.

Minor cosmetic comments and typos.
- table 1 overflows to margins
- table 2 and 3 overlap
- 3.2 paragraph 1 - "each conditioning latent encode", it's unclear what this means. Might want to say feature instead of encode
- 3.2 paragraph 3 - "convolution encoder" -> convolutional encoder

---

> ### Author Response · Authors · 2023-12-17
> **Response to Reviewer qD6L Part 1**
>
> Thank you for your constructive comments and feedback!  We have revised the paper with new experiments and discussions based on your feedback, and we respond to individual points below.
>
> 1. *Additional explanation in the draft about what is the reasoning behind the design decisions of how the training/testing data is generated w.r.t the points discussed above?*
>
>     The reasoning behind the design decision of the dataset is based on the observation that SOTA unsupervised object-centric 3D learning method uORF fails to apply to more complex scenes. uORF works well on scenes with a single object category (i.e. chair) and uniformly-colored object instances without textures. However, it fails to work on scenes with more diversity in object categories, shapes, and textures, which is one of the most prominent sources of complexities and challenges in working with real-world scenarios. Therefore, the purpose of the dataset design is to drastically increase object complexity by including 231 object categories (2000 unique object models in total) in the Playroom dataset. This dataset is substantially more complex than the one used in the previous approach and serves as a strong test for how a model could handle scenes with diverse object textures and shapes. Nevertheless, we agree with the reviewer that including other real-world challenges would strengthen our work. In the following, we address comments regarding challenges other than object complexity.
>
>
>
> 2. *One of the strengths of synthetic data is that it allows us to gradually make testing data more challenging across different axes (e.g. increasing the number of objects, increasing the amount of occlusion, using completely novel environments). Adding additional results that take advantage of this would greatly improve the paper, and help us understand how well MORF actually works, and what should be the goals of future work.*
>
>     We have included additional experiments and results in the papers according to the reviewer’s suggestions, including scenes with more diverse background object environments, scenes with an arbitrary number of objects, and scenes with more realistic object poses. Please see points 3 - 7 below for more discussions of the results, in response to the reviewer’s suggestions.
>
> 3. *in Playroom there are only 3 environments, and they're used both for training and testing. MultiShapeNet has one environment.*
>
>     We generate an additional dataset called Playroom-diverse, which extends Playroom dataset to include more diverse background environments. Playroom dataset uses 3 environments, and MultiShapeNet contains a single environment. Playroom-diverse contains 30 unique background environments. More details and qualitative examples of the datasets can be found in A.1 of the appendix. We generate an additional 30,000 scenes for finetuning and 600 scenes for testing. We report both quantitative and qualitative results of MORF and other baseline methods in Table 1 and Figure 3. We also include more qualitative results in Figure 12 of the appendix. We demonstrate that MORF is still better than the baseline methods on the Playroom-diverse dataset with higher complexity. We include the novel view synthesis metrics of MORF and other baseline methods on the Playroom-diverse dataset below:
>
>
>     | Method              | LPIPS &darr; | SSIM &uarr;| PSNR &uarr; |
>     | :---------------- | :------: | :------: |:------:|
>     | uORF              | 0.553 | 0.547 | 18.1|
>     | PixelNeRF      | 0.357 | 0.650|  20.4|
>     | SRT                | 0.451 | 0.582 | 18.7|
>     | MORF-nopix | 0.466 | 0.602 | 18.4 |
>     | MORF (ours)    | **0.248** |  **0.675** |  **21.3** |
>
> 4. *In Playroom, there are 100 novel objects, but it's not clear whether these are from novel categories or not, or how different they are from the objects used in the training set. In MultiShapeNet it's not clear whether the training/validation objects are distinct.*
>
>     Playroom-novel contains a mixture of both novel object instances from categories seen during training, as well as object instances from novel categories. Object instances from novel categories are randomly sampled from 36 held-out categories. We have updated the dataset description in the paper.
>
> 5. *there are at most 4 objects in each scene*
>
>      To test the model’s ability to generalize to a different number of objects, we generate additional testing scenes with a random number of objects ranging from 1 to 6. With object-centricity and the compositional nature of the novel view synthesis, MORF can generalize well to test scenes with a different number of objects. Figure 14 shows the qualitative results of the most challenging scenarios with 4-6 objects in a scene.

---

> > ### Author Response · Authors · 2023-12-17
> > **Response to Reviewer qD6L Part 2**
> >
> > 6. *the objects appear to be well-spaced, and the camera is at a fixed height. As a result, it appears that object-object occlusion doesn't happen very often, which is unrealistic.*
> >
> >     We show qualitative examples of partial object occlusion in the Playroom validation set in the first two rows of Figure 15. To further test the ability of MORF in scenes with more object occlusion, we create more challenging test scenes with objects placed closer to each other, as shown in the last two rows of Figure 15. Although the geometric reconstruction of occluded objects is not as sharp as compared to the unoccluded objects in the input view, MORF is generally capable of handling partial object occlusion and performing approximate amodal completion of the occluded objects.
> >
> > 7. *Objects have similar scales and are placed in canonical poses. For example in the real world, an apple is much smaller than a table or a bowl and is usually found on top of a table or inside a bowl.*
> >
> >     Playroom dataset simulates an infant playing environment with toys placed on the floor. We acknowledge that objects are often stacked on top of one another in real-world scenarios. To further improve the dataset, we generate additional scenes with three types of stacking configurations commonly seen in real life: (a) a cup on top of a coaster, (b) a food item on the plate, and (c) an object on top of a microwave. We generate 1000 scenes for each stacking configuration and fine-tune MORF on the additional scenes. Figure 16 shows the qualitative results of novel view syntheses on held-out scenes and demonstrates that MORF is still able to accurately synthesize novel views of scenes with stacking objects.
> >
> > 8. *The camera is at a fixed height.*
> >
> >     MORE aims to take in a monocular image and reconstruct novel views for scenes that are not seen during training. To a monocular camera, there is an inherent issue of depth ambiguity: a small object close to the camera and a large object at a distance appear identical, despite representing drastically different scenes. As a result of scale ambiguity in the input view, it causes ambiguity in novel view synthesis, preventing a meaningful quantitative evaluation on novel view synthesis or novel view segmentation. Thus, to allow quantitative evaluation, we fix the camera height to remove the depth scale ambiguity.
> >
> > 9. *Costimetic comments and typos*
> >
> >     We thank the reviewer for the note. We have updated the draft accordingly to fix all typos.

---

### Review · Reviewer_hAH5 · 2023-11-12

**Summary Of Contributions:**

This paper is about how to extract object-level radiance fields so that objects can be later removed, inserted, and re-arranged in a scene.
They propose Movable Object Radiance Fields. The method consists of object segmentation through optical flow and radiance field extraction conditioned on object segmentation. A novel dataset is proposed.
The paper belongs to the field of unsupervised scene decomposition.

For 2D object segmentation, the paper uses EISEN which learns similarities using the RAFT optical flow method. EISEN uses pairs during training but only single images during inference. Then, given predicted object masks and an RGB image,  global and local features are computed that are fed as conditioning variables to two radiance fields, one for the background and one for each object in the foreground.

**Audience:**

Yes

**Broader Impact Concerns:**

None.

**Claims And Evidence:**

No

**Requested Changes:**

0. Respond to the main weakness above.

1. The abstract and the introduction are incomprehensible. It is unclear which problem the authors address regarding input and output during inference and training.

2. "While supervised learning methods have shown promise in learning 3D object representations (such as neural radiance fields) from images, they rely on annotations"

This is really confusing. NeRFs do not use any annotation and the only supervision signal is the views themselves.

3. Fig.2: Please provide an additional figure about what exactly happens during inference.

4. Authors should clarify the dimensions of the feature map and over which dimension pooling is happening (3.2) to obtain $s_i$ from $f_i$.

5. It is not clear what is the meaning of the GRU. An ablation would be illuminating.

5. During the method description it is not clear whether multiple views are used for the NeRF optimization.

6. During inference it is not clear, how an object radiance field is subtracted from or added to the scene.

7. It was not clear whether EISEN is frozen during training.

8. There is no quantitative evaluation in real images.

**Strengths And Weaknesses:**

STRENGTHS

The problem of 3D scene decomposition from a single image is a challenging problem. Instead of using full 3D supervision for metric reconstruction and semantics (the way for example one would use Scannet), the authors use only supervision for 2D semantics and then use the self-supervision of NeRFs to extract one radiance field per object. The authors evaluate on novel view synthesis, mesh reconstruction, and scene editing.

WEAKNESSES

The main weakness of the paper is the lack of clarity in the method and what exactly happens during training and during inference. The fundamental challenge on 3D scene decomposition is that segmentation and reconstruction is a chicken and egg problem: If one knows the segmentation, then reconstruction is feasible (what the paper here follows) but, on the other hand, segmentation should depend on
the geometrically consistent reconstruction. Unfortunately, the current scheme does not allow the segmentation to be informed by the reconstruction. This would require a recurrent unit taking advantage of reconstruction/rendering inconsistencies.

---

> ### Author Response · Authors · 2024-01-19
> **Response to Reviewer hAH5 Part 1**
>
> Thank you for your constructive comments and feedback!  We have revised the paper with new experiments and discussions based on your feedback, and we respond to individual points below.
>
> 0. *Respond to the main weakness above*
>
>     We note that a fundamental bottleneck of existing unsupervised 3D scene decomposition methods is the failure to discover complex objects due to the lack of object priors. For example, the state-of-the-art method, uORF, works well on scenes with a single object category (e.g., chair) and uniformly-colored object instances without textures. However, it fails to decompose scenes with more diversity in object categories and textures due to inaccurate 2D segmentations, which in turns result in inaccurate 3D radiance fields. As a result, existing methods do not allow segmentation to be informed by reconstruction. To resolve this dilemma, the goal of this paper is to inject object prior using an unsupervised object segmentation method called EISEN. Since the 2D segmentations discovered by EISEN are highly accurate, we did not find it useful to take advantage of reconstruction inconsistencies for further improving segmentations.
>
> 1. *The abstract and the introduction are incomprehensible. It is unclear which problem the authors address regarding input and output during inference and training*
>
>     This paper solves the important problem of unsupervised object-centric 3D representation learning from a single image of unseen scenes. During both training and inference, MORF takes in a single image of a scene as input, and outputs the inferred 3D object and background radiance fields. During training, MORF reconstructs multiple novel views via compositional rendering and is supervised by reconstruction losses between the reconstructed views and the ground-truth views. During inference, MORF takes a single view of a new scene, and infers object and background radiance fields in a single forward pass.
>
>
> 2. *While supervised learning methods have shown promise in learning 3D object representations (such as neural radiance fields) from images, they rely on annotations*
>
>      We clarify that NeRF is referred to in this sentence as an example of 3D object representations, not an example of supervised learning methods. “Supervised learning methods" here refer to works included in the citation, such as Neural Scene Graph [1] and Panoptic Neural Fields [2]. Neural Scene Graph relies on annotations of object poses, whereas Panoptic Neural Fields rely on a supervised segmenter trained using ground-truth semantic segmentations. We have updated the draft for better clarity.
>
> 3. *Fig.2: Please provide an additional figure about what exactly happens during inference.*
>
>     We provide an additional figure about the inference procedure (Figure 10 in Supplement A.4) . MORF takes a single view of a new scene as input and infers object and background radiance fields in a single forward pass. Given an arbitrary camera pose, a novel view can be synthesized via compositional rendering.
>
> 4. *Authors should clarify the dimensions of the feature map and over which dimension pooling is happening (3.2) to obtain $s_i$ from $f_i$*
>
>     The feature maps $f_i$ have dimensions [B, C, H’, W’], where C=64, and H’=W’=64, as specified in Table 6 of the supplementary. The pooling is happening over the 2D spatial dimensions, which obtain $s_i$ of shape [B, C] from $f_i$ of shape [B, C, H’, W’].
>
> 5. *It is not clear what is the meaning of the GRU. An ablation would be illuminating.*
>
>     We follow the previous method Slot Attention [3] in using the Gated Recurrent Unit (GRU) for computing object latents. In the Slot Attention paper (Figure 11), the authors conduct an ablation experiment and demonstrate that using GRU in learning the update function for the slots as opposed to simply taking the output of the weighted mean as the next value for the slots yields noticeable improvement.
>
> \
>         [1] Neural Scene Graphs for Dynamic Scenes, Ost et. al., 2021\
>         [2] Panoptic Neural Fields: A Semantic Object-Aware Neural Scene Representation, Kundu et. al., 2022\
>         [3] Object-Centric Learning with Slot Attention, Locatello et. al., 2020

---

> ### Author Response · Authors · 2024-01-19
> **Response to Reviewer hAH5 Part 2**
>
> 6. *During the method description it is not clear whether multiple views are used for the NeRF optimization.*
>
>     Multiple views are used to train MORF, as specified in the first sentence of section 3.4. We use 4 views per scene in the training dataset. During training, we randomly select a single view as input and use the other 3 views in computing reconstruction loss for MORF learning.
>
> 7. *During inference it is not clear, how an object radiance field is subtracted from or added to the scene.*
>
>     To compose object radiance fields into the holistic scene, we model a scene by a composition of multiple object radiance fields. The scene is represented by the compositional scene, i.e., a density-weighted mean to combine all objects radiance fields. These object radiance fields can be specified by a list of density {$\sigma_1$, …, $\sigma_n$} and color {$c_1$, …, $c_n$} at any 3D location, where *n* is the number of objects. To add or subtract an object radiance field, we can simply add or remove its corresponding density $\sigma_i$ and color $c_i$ from the list when computing density-weighted mean.
>
> 8. *It was not clear whether EISEN is frozen during training.*
>
>      EISEN is pre-trained and frozen during the training of MORF, as specified at the beginning of the second paragraph in section 3.2.
>
> 9. *There is no quantitative evaluation in real images.*
>
>     We train MORF on a real-world dataset including scenes with four plant pots or vases on a tabletop decorated with tablecloths. We gathered 9 plant pots, 8 vases, and 6 tablecloths and captured 745 scenes for training and 140 for evaluation, each with 3 images from different poses. We include the quantitative evaluation in the table below. We also include qualitative novel view synthesis results on the testing real-world scenes in Figure 19 of Supplement A.5. We show that MORF is able to synthesize the novel view accurately, whereas uORF fails to decompose the scenes, resulting in inaccurate 3D object radiance fields.
>
>
>
>      | Method              | LPIPS &darr; | SSIM &uarr;| PSNR &uarr; |
>      | :---------------- | :------: | :------: |:------:|
>      | uORF              | 0.336 | 0.577 | 19.8|
>      | MORF-nopix | 0.323 | 0.580 | 19.9 |
>      | MORF (ours)    | **0.064** |  **0.903** |  **31.0** |

---

### Decision · Action_Editor_YPZc · 2024-02-09

**Recommendation:** Accept with minor revision

**Comment:**

After revision the reviews were borderline and the AE read the paper and the reviews to understand the issues. On the one hand, the reviewers thought that the problem was timely, worth tackling and thought that the method was interesting. On the other hand, there were residual concerns about novelty of the contributions and the real-worldness of the results. These concerns were shared by multiple reviewers.

Given the mixed recommendations, the AE has considered the identified weaknesses of the current paper. The AC thinks that the paper's strengths outweigh its weaknesses.
1. *Novelty* TMLR, as a journal, has a strong focus on well-supported claims as opposed to novelty. While the AE does understand the origin of the reviewer concerns about novelty, novelty (in the CVPR/NeurIPS/ICML-sense) by itself carries less strength for TMLR decisions as compared to whether the claims are demonstrated and the method is of interest.
2. *Real-world results* The AE understands the concern about real world experiments -- some things transfer well from rendered scenes and others do not. However, the AE thinks that the additional real-world results have demonstrated the claims sufficiently well for the purposes of this paper and its claims (although improving the realness of the modeling may be important for tackling more complex scenes with photometric interaction as hAH5 points out).

At the same time, the AE does think a small modification would be helpful and would alleviate the concerns of the reviewers about the real world data. The paper would be improved if the final claims-oriented paragraph of the introduction and potentially the abstract were more specific. For instance,  "complex 3D scenes with textured objects from diverse unseen categories" in the introduction would be improved by specifying the datasets on which this is demonstrated. The AE doesn't want to force the authors to write this and it is up to the authors, but the AE would encourage the authors to consider this sort of modification (and is giving "minor revisions" to enable this).

**Audience:**

The paper, its topic, and its method are of interest to members of the TMLR community.

**Claims And Evidence:**

The claims and evidence are supported by the current paper. Although there are some concerns about the realness of some of the data (that the AE also shares), the paper overall justifies its current claims. This is discussed in more depth in the comment.